# Destructive disinfection of infected brood prevents systemic disease spread in ant colonies

Christopher D Pull[1†]*, Line V Ugelvig[1‡], Florian Wiesenhofer[1§], Anna V Grasse[1], Simon Tragust[1,2#], Thomas Schmitt[3], Mark JF Brown[4], Sylvia Cremer[1]*

[1]IST Austria (Institute of Science and Technology Austria), Klosterneuburg, Austria; [2]Evolution, Genetics and Behaviour, University of Regensburg, Regensburg, Germany; [3]Department of Animal Ecology and Tropical Biology, University of Würzburg, Würzburg, Germany; [4]School of Biological Sciences, Royal Holloway University of London, Egham, United Kingdom

**Abstract** In social groups, infections have the potential to spread rapidly and cause disease outbreaks. Here, we show that in a social insect, the ant *Lasius neglectus*, the negative consequences of fungal infections (*Metarhizium brunneum*) can be mitigated by employing an efficient multicomponent behaviour, termed destructive disinfection, which prevents further spread of the disease through the colony. Ants specifically target infected pupae during the pathogen's non-contagious incubation period, utilising chemical 'sickness cues' emitted by pupae. They then remove the pupal cocoon, perforate its cuticle and administer antimicrobial poison, which enters the body and prevents pathogen replication from the inside out. Like the immune system of a metazoan body that specifically targets and eliminates infected cells, ants destroy infected brood to stop the pathogen completing its lifecycle, thus protecting the rest of the colony. Hence, in an analogous fashion, the same principles of disease defence apply at different levels of biological organisation.
DOI: https://doi.org/10.7554/eLife.32073.001

**\*For correspondence:**
chris.pull@rhul.ac.uk (CDP);
sylvia.cremer@ist.ac.at (SC)

**Present address:** †School of Biological Sciences, Royal Holloway University of London, Egham, United Kingdom; ‡Centre for Social Evolution, Department of Biology, University of Copenhagen, Copenhagen, Denmark; §Department of Pediatrics and Adolescent Medicine, Medical University of Vienna, Vienna, Austria; #General Zoology/ Institute of Biology, University of Halle, Halle/Saale, Germany

**Competing interests:** The authors declare that no competing interests exist.

## Introduction

Pathogen replication and transmission from infectious to susceptible hosts is key to the success of contagious diseases (*Schmid-Hempel, 2011*). Social animals are therefore expected to experience a greater risk of disease outbreaks than solitary species, because their higher number of within-group interactions will promote pathogen spread (*Nunn and Altizer, 2006*; *Schmid-Hempel, 2017*; *Alexander, 1974*). As a consequence, evolutionary immunology predicts that traits mitigating this cost, such as detecting sick conspecifics and using that information to prevent self-infection, should have been selected for in group-living animals as an essential adaptation to social life (*Hamilton, 1987*; *Ezenwa et al., 2016*).

Social animals, including lobsters, tadpoles, mice and mandrills, can use conspicuous disease-associated changes in the physical appearance, behaviour and chemical odour of conspecifics to identify sick group members (*Shakhar and Shakhar, 2015*; *Arakawa et al., 2011*; *Lopes, 2014*; *Bozza, 2015*). Upon detection, healthy animals usually respond by interacting with sick conspecifics less or avoiding them completely (*Poirotte et al., 2017*; *Kiesecker et al., 1999*; *Behringer et al., 2006*; *Anderson and Behringer, 2013*). In addition, they may prophylactically increase the expression of their immune defences in preparation for a potential immune challenge (*Hernández López et al., 2017*), and a similar phenomenon is observed in animals and plants in response to chemicals released by wounded conspecifics (*Heil and Silva Bueno, 2007*; *Peuß et al., 2015*). Because there

**eLife digest** Ants live in crowded societies where disease can spread rapidly and take a heavy toll on the community. Ants have a number of ways to prevent these outbreaks before they become a problem. Like many other social species, they practice good hygiene and groom nest mates that have picked up a pathogen, which helps them to recover and to reduce the likelihood of the disease spreading.

Unlike other social species, ants appear to have evolved collective disease defence, or social immunity, because their colonies behave like a 'superorganism', in which the society behaves much like a single organism would. Like an individual animal that has an infection, the colony needs to be able to eliminate infections collectively when a nest mate falls ill, to prevent the disease from spreading.

To understand how an ant colony protects itself when the care fails and a colony member contracts a lethal infection, Pull et al. infected the brood of the invasive garden ant with a common soil fungus. Using a combination of chemical analyses and behavioural observations, it was shown that the infected pupae emitted a chemical cue, which the tending ants could detect. Using a microscopic camera, Pull et al. found that when the ants sensed the cue, they would unpack the infected pupae from their cocoons and bite them. They then sprayed them with an antiseptic poison, which entered the hole in the pupae's body, killing both the pupae and the fungus inside, before it had a chance to spread.

This process of destructive disinfection may seem like a large sacrifice, but it helps to protect the rest of the colony from a fungus that could lead to much greater damage. The tending ants were acting within the superorganism of the colony much like immune cells act within an individual's body – honing in on infected cells and destroying them before the pathogen can spread to other cells. This suggests that the ability to detect and destroy harmful elements was necessary for both the evolution of multicellular organisms, and from single animals to superorganisms.

DOI: https://doi.org/10.7554/eLife.32073.002

are downsides to being socially excluded, sick animals may, under some circumstances, attempt to hide their illness (*Lopes, 2014*; *Lopes et al., 2012*). However, signalling infection to others should be adaptive if (i) this elicits care from conspecifics that improves the chance of recovery (*Hart, 1990*) or (ii) if the infection is otherwise likely to spread and infect kin (*Shakhar and Shakhar, 2015*). This is because individuals can gain indirect fitness by enhancing the propagation of shared genes, present in relatives, into the next generation (*Hamilton, 1964*). Hence, in a closely related social group, an animal that warns its relatives if it is sick is likely to have a greater inclusive fitness than an animal that does not, since fewer of its kin will fall sick and suffer reductions in fitness (*Shakhar and Shakhar, 2015*). Hence, in closely related social groups, there may be selection for both the detection of illnesses by healthy group members (*Curtis, 2014*) and the advertisement of an animal's disease status by sick individuals themselves (*Shakhar and Shakhar, 2015*).

In the social insects (termites and ants and the social bees and wasps), colonies are typically single families, comprised of a queen and her daughters, the workers (*Queller and Strassmann, 2003*). They typically have an irreversible reproductive division of labour, with the two castes being highly interdependent: the queens are morphologically specialised for reproduction and cannot survive without the assistance of the workers; conversely, the workers cannot reproduce, but gain fitness indirectly by raising the queen's offspring (*Queller and Strassmann, 2003*). Consequently, social insect societies have become indivisible, reproductive units, where natural selection acts on the colony instead of its individual members (*Bourke, 2011*; *West et al., 2015*). This has parallels to the evolution of complex multicellular organisms, that is metazoan bodies, where sterile somatic tissue and germ line cells form an indivisible reproducing body. Hence, social insect colonies are often termed 'superorganisms' and their emergence is considered a major evolutionary transition (*Bourke, 2011*; *West et al., 2015*; *Wheeler, 1911*; *Boomsma and Gawne, 2017*). Since evolution favours the survival of the colony over its members, selection has resulted in a plethora of cooperative and altruistic traits that workers perform to protect the colony from harm (*Hamilton, 1987*; *Bourke, 2011*; *Cremer et al., 2007*; *Shorter and Rueppell, 2012*; *Rosengaus et al., 2011*;

*Boomsma et al., 2005*; *Cremer et al., 2017*). In particular, social insects have evolved physiological and behavioural adaptations that limit the colony-level impact of infectious diseases, which could otherwise spread easily due to the intimate social interactions between colony members (*Cremer et al., 2007*; *Rosengaus et al., 2011*; *Boomsma et al., 2005*; *Cremer et al., 2017*; *Stroeymeyt et al., 2014*; *Meunier, 2015*). These defences are performed collectively by the workers to form an emergent layer of protection known as social immunity that, like the immune system of a body, protects the colony from invading pathogens (*Cremer et al., 2007*; *Rosengaus et al., 2011*; *Cremer et al., 2017*; *Cremer and Sixt, 2009*).

Our understanding of how social immunity functions is based mostly on the behaviours social insects perform to prevent infection in contaminated colony members, referred to as sanitary care (*Cremer et al., 2007*; *Rosengaus et al., 2011*; *Tragust et al., 2013a*; *Wilson-Rich et al., 2009*). In ants, sanitary care involves grooming and the use of antimicrobial secretions to mechanically remove and chemically disinfect pathogens, reducing the likelihood that pathogen exposure leads to the development of an infection (*Tragust et al., 2013a*; *Graystock and Hughes, 2011*). During sanitary care, protection of the colony is aligned with the protection of the individual colony member. In contrast, this is not the case if an individual is infected with a contagious disease, since they risk infecting the rest of the colony if the pathogen spreads (*Cremer et al., 2017*). Hence, social immunity is characterised by a care-kill dichotomy, where colony members should be cared for when possible but sacrificed if necessary, both of which benefit the colony (*Cremer et al., 2017*; *Cremer and Sixt, 2009*). Although the kill-component is an unique feature of social immunity that is not present in the disease defence repertoire of other forms of sociality (e.g. non-superorganismal family groups, communal breeders or aggregations [*Cremer et al., 2017*]), it is rarely studied in comparison to the care-component. Hence, what happens when sanitary care fails and a pathogen successfully infects a colony member, with the consequent potential to create an epidemic, remains poorly understood (but see [*Rothenbuhler, 1964*; *Ugelvig et al., 2010*; *Tragust et al., 2013b*]). To effectively protect the colony, we predict that infections should be detected quickly and accurately by nestmates, in order to overcome the pathogen before it has time to replicate and produce propagules that can infect others. Once detected, the colony should respond by eliminating the infection as effectively as possible, by preventing any further opportunities for disease transmission. This is especially pertinent in the ants and termites, because their sedentary and territorial lifestyle makes it likely infectious corpses are encountered again, even if they are removed from the colony (*Boomsma et al., 2005*; *Cremer et al., 2017*; *Schmid-Hempel, 1998*). For example, previous studies have shown that fungus-infected ants become highly contagious to nestmates after death and can cause epidemics that result in colony collapse (*Hughes et al., 2002*; *Loreto and Hughes, 2016*).

To address the above gaps in our knowledge of social immunity, we investigated how ants detect and respond to infections. To that end, we exposed the immobile brood of the invasive garden ant, *Lasius neglectus,* to a generalist fungal pathogen, *Metarhizium brunneum*. When the infectious conidiospores of this fungus come into contact with insect cuticle, they attach, germinate and penetrate the host cuticle within 24–48 hr to cause internal infections (*Vestergaard et al., 1999*; *Walker and Hughes, 2009*). During a short, non-infectious incubation period following infection, the fungus goes through a single-cell blastospore stage, which lasts 1–2 days. Once the host has died, the fungus enters a saprophytic mycelial phase, then grows out of the corpse, 1–3 days later, releasing millions of new infectious conidiospores, in a process called sporulation (*Hughes et al., 2002*; *Deacon, 2006*). Previous work found that brood infected with *Metarhizium* is removed from the brood chamber (*Ugelvig et al., 2010*; *Tragust et al., 2013b*), however, it is unknown how the ants then respond to the infection. Hence, in this study, we performed a series of behavioural and chemical experiments to test how ants detect and prevent infected brood from causing a systemic colony infection.

## Results

### Destructive disinfection of lethally infected pupae

We exposed ant pupae to one of either three dosages of *Metarhizium* conidiospores or a sham control. We observed that ants tending pathogen-exposed pupae prematurely removed the pupae from their cocoons in a behaviour we termed 'unpacking', whereas control pupae were left

cocooned (*Figure 1A–B*, *Video 1*; Cox proportional hazards regression: likelihood ratio test (LR) $\chi^2$ = 55.48, df = 3, p<0.001; hazard ratios (x greater chance of unpacking compared to control): low dose = 18, medium = 53, high = 111; post-hoc comparisons: control vs. low, p=0.004; low vs. medium, p=0.006; medium vs. high = 0.024; all others, p=0.001). Unpacking occurred between 2 and 10 days after pathogen exposure, but sooner and more frequently at higher conidiospore dosages (*Figure 1B*). As unpacking was a belated response to pathogen exposure and we were unable to remove any conidiospores from the majority of the cocoons or the unpacked pupae (number of colony forming units [mean ± 95% CIs]: cocoons = 0.6 ± 0.9; pupae = 0.1 ± 0.35; *Figure 1—figure supplement 1*), we concluded that the ants were not performing unpacking to simply dispose of contaminated cocoons. Instead, we postulated that unpacking was a response to successful infection of the pupae. At the time of unpacking, the majority of pupae were still alive (i.e. had an active dorsal aorta pulse; *Figure 1—figure supplement 2*) and fungal outgrowth had not yet occurred (*Figure 1F*). Hence, to test if the ants were reacting to early-stage infections, we removed both

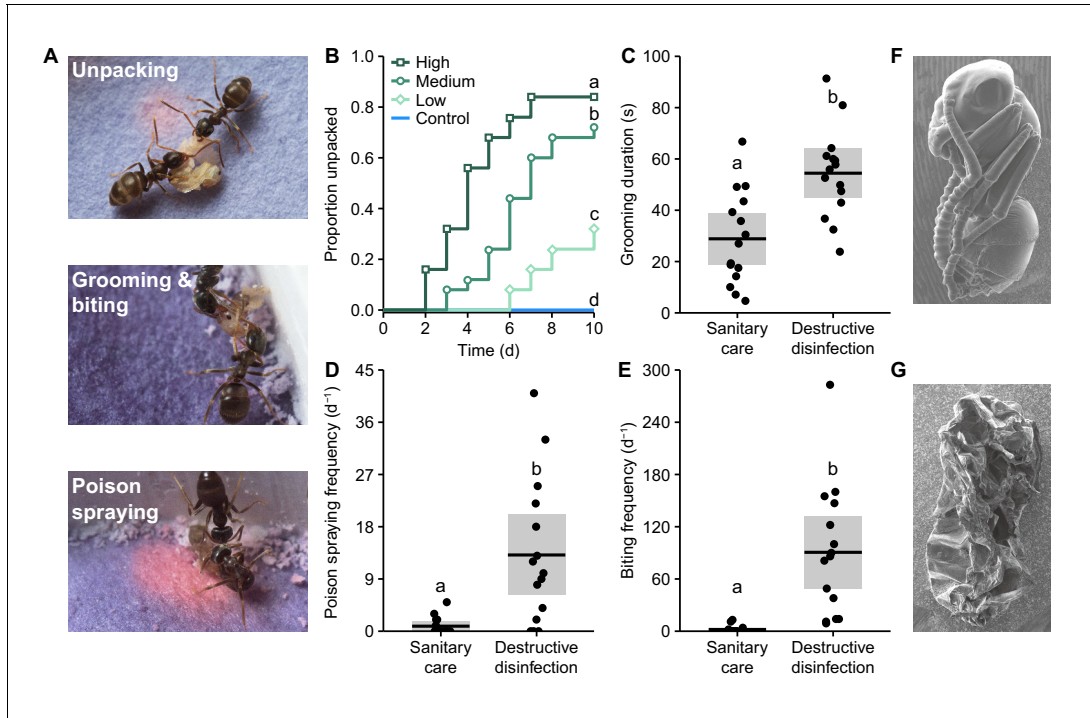

**Figure 1.** Ants perform destructive disinfection in response to lethal fungal infections of pupae. (A) Destructive disinfection starts with the unpacking of pupae from their cocoons and is followed by grooming, biting and poison spraying (ants housed on blue pH-sensitive paper to visualise acidic poison spraying, which shows up pink). (B) Unpacking occurred when pupae were exposed to fungal conidiospores and was dose-dependent, happening sooner and in higher amounts as the dose of conidiospores increased (letters denote groups that differ significantly in post-hoc comparisons [model revelling; p<0.05]). (C–E) Comparison of the ants' behaviour between sanitary care and destructive disinfection. Destructive disinfection is characterised by increases in grooming duration, poison spraying frequency and biting frequency (all data points displayed; lines ± shaded boxes show mean ± 95% confidence intervals [CI]; letters denote groups that differ significantly in logistic regressions [p<0.05]). (F) Scanning electron micrographs (SEM) of an asymptomatic infected pupa immediately after unpacking, and (G) of a destructively disinfected pupa 24 hr later.

DOI: https://doi.org/10.7554/eLife.32073.003

The following figure supplements are available for figure 1:

**Figure supplement 1.** Conidiospore load on pupae.
DOI: https://doi.org/10.7554/eLife.32073.004

**Figure supplement 2.** Unpacked pupae are killed by destructive disinfection.
DOI: https://doi.org/10.7554/eLife.32073.005

**Figure supplement 3.** Unpacked pupae are infected.
DOI: https://doi.org/10.7554/eLife.32073.006

**Figure supplement 4.** Destructive disinfection reduces pupa pH.
DOI: https://doi.org/10.7554/eLife.32073.007

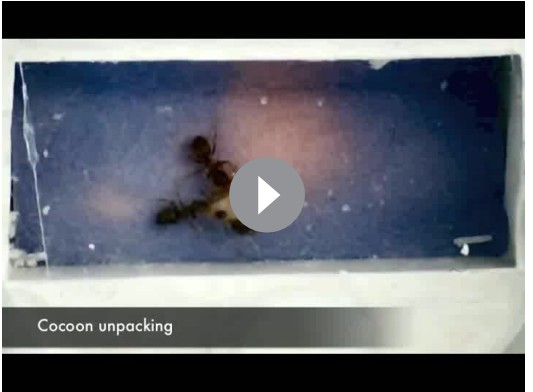

**Video 1.** Ants performing destructive disinfection of an infected pupa. Video shows *Lasius neglectus* ants performing destructive disinfection towards a *Metarhizium*-infected pupa. Video playback is 8 x normal speed.
DOI: https://doi.org/10.7554/eLife.32073.008

unpacked and non-unpacked pathogen-exposed cocooned pupae from the ants and incubated them under optimal conditions for fungal outgrowth. We found that, on average across the conidiospore dosages, 85% of unpacked pupae harboured infections that sporulated in the absence of the ants. In contrast, only 25% of non-unpacked pupae were infected (*Figure 1—figure supplement 3*; generalised linear model [GLM]: overall LR $\chi^2$ = 26.48, df = 5, p<0.001; cocooned vs. unpacked pupae: LR $\chi^2$ = 18.5, df = 1, p=0.001; conidiospore dose: LR $\chi^2$ = 0.42, df = 2, p=0.81). We therefore concluded that the ants were detecting and unpacking pupae with lethal infections during the asymptomatic incubation period of the pathogen's lifecycle. At this time point, the fungus growing inside the pupae is non-infectious and essentially no viable conidiospores are leftover from the pathogen exposure on their cuticle (*Figure 1—figure supplement 1*), so there is very little risk of the ants contracting the disease whilst unpacking the pupae.

Next, we filmed ants presented with pathogen-exposed pupae and compared their behaviour before and after unpacking. Prior to unpacking, we observed the typical sanitary care behaviours reported in previous studies (*Graystock and Hughes, 2011*; *Tragust et al., 2013a*; *Hughes et al., 2002*; *Reber et al., 2011*; *Okuno et al., 2012*). Namely, the ants groomed the pupae (*Figure 1C*), which has the dual function of removing the conidiospores and applying the ants' antimicrobial poison (*Tragust et al., 2013a*). In *L. neglectus*, the poison is mostly formic acid and is emitted from the acidopore at the abdominal tip, where the ants actively suck it up and transiently store it in their mouths until application during grooming. Additionally, the ants can spray their poison directly from the acidopore; yet, this behaviour is rarely expressed during sanitary care (about once every 28 hr; *Figure 1D*[*Tragust et al., 2013a*]). However, after unpacking, we observed a set of behaviours markedly different to sanitary care (*Figure 1A*, *Video 1*). The ants sprayed the pupae with poison from their acidopore approx. 15-times more frequently than during sanitary care (~13 times/d; *Figure 1D*; generalised linear mixed model [GLMM]: LR $\chi^2$ = 17.04, df = 1, p<0.001), and grooming duration doubled (*Figure 1C*; linear mixed effects regression [LMER]: LR $\chi^2$ = 145.26, df = 1, p<0.001). Given that there was no fungus to remove at the time of unpacking, the increase in grooming probably functioned solely to apply poison from the oral store (*Tragust et al., 2013a*). Furthermore, the ants repeatedly bit the pupae to make perforations in their cuticles and to remove their limbs (*Figure 1E*; GLMM: LR $\chi^2$ = 39.44, df = 1, p<0.001). Together these three behaviours resulted in the death of the pupae and left their corpses heavily damaged and coated in the ants' poison (*Figure 1G*, *Figure 1—figure supplements 2* and *4*). Accordingly, we named the combination of unpacking, grooming, poison spraying and biting 'destructive disinfection', and performed a series of experiments to determine its function.

## Chemical detection of internal infections

Firstly, we wanted to know how the ants identify internal infections during the pathogen's non-contagious incubation period, when pupae were still alive and showed no external signs of disease. As ants use chemical compounds on their cuticles to communicate complex physiological information to nestmates (*Leonhardt et al., 2016*), we speculated that infected pupae may produce chemical sickness cues. We washed infected pupae in pentane to reduce the abundance of their cuticular hydrocarbons (CHCs). When pentane-washed pupae were presented to ants, there was a 72% reduction in unpacking compared to both non- and water-washed infected pupae (*Figure 2A*; GLM: LR $\chi^2$ = 12.2, df = 2, p=0.002; Tukey post hoc comparisons: water-washed vs. non-washed, p=0.79; all others, p=0.009). As pentane-washed pupae had lower abundances of CHCs (*Figure 2—figure*

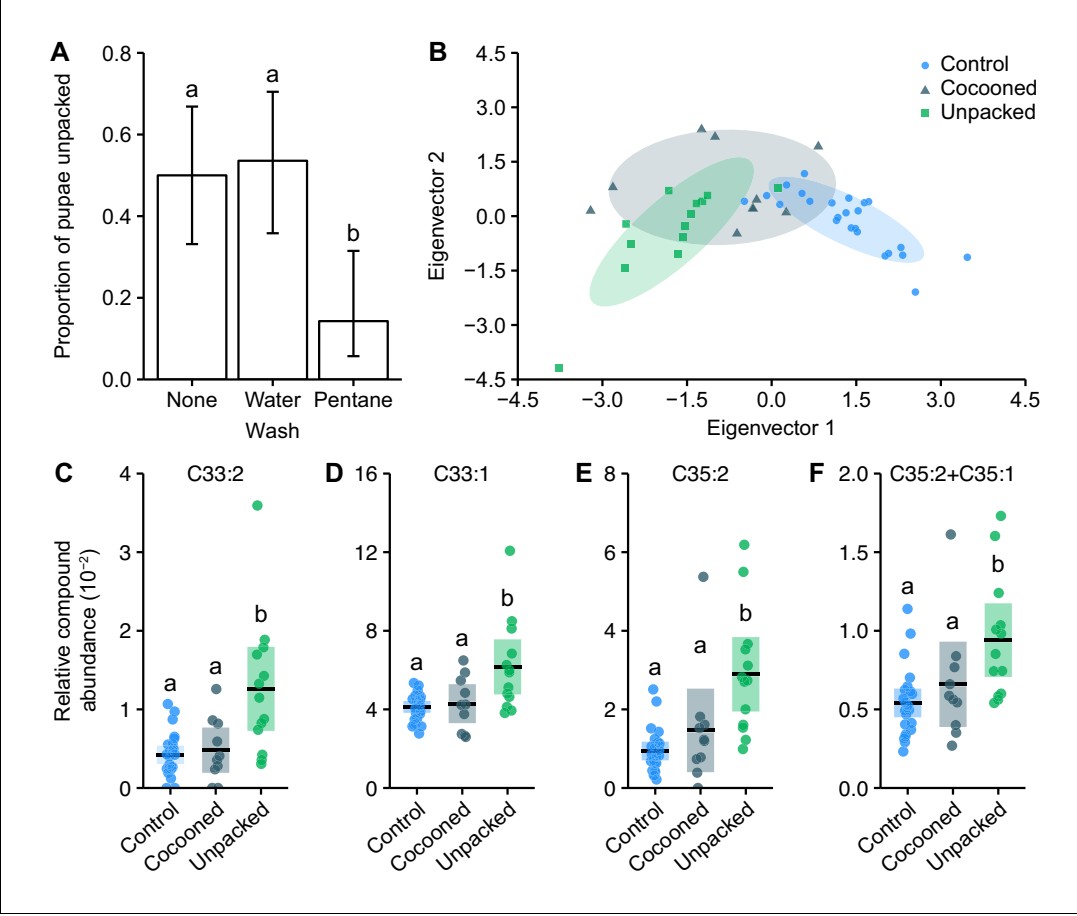

**Figure 2.** Destructive disinfection is induced by changes in the chemical profile of infected pupae. (A) Pupae washed in pentane solvent to reduce the abundance of their cuticular hydrocarbons (CHCs) were unpacked less than unwashed or water-washed pupae (positive and handling controls, respectively; error bars show ± 95% CI; letters specify significant Tukey post hoc comparisons [p<0.05]). (B) Unpacked pathogen-exposed pupae have distinct chemical profiles compared to sham-treated control pupae. Pathogen-exposed pupae that were not unpacked (cocooned group) have intermediate profiles (axes show discriminant analysis of principle components eigenvectors). (C–F) The four CHCs with higher relative abundances on unpacked pupae compared to both control and cocooned pupae: (C) Tritriacontadiene, C33:2 (D), Tritriacontene, C33:1 (E), Pentatriacontadiene, C35:2 (F) co-eluting Pentatriacontadiene and Pentatriacontene, C35:2 + C35:1 (all data points displayed; line ± shaded box show mean ± 95% CI; letters specify groups that differ significantly in KW test post hoc comparisons [p<0.05]).

DOI: https://doi.org/10.7554/eLife.32073.009

The following figure supplements are available for figure 2:

**Figure supplement 1.** Total abundance of cuticular hydrocarbons (CHCs) on pupae.
DOI: https://doi.org/10.7554/eLife.32073.010

**Figure supplement 2.** The cuticular hydrocarbon (CHC) profiles of unpacked and control pupae.
DOI: https://doi.org/10.7554/eLife.32073.011

**Figure supplement 3.** Change in immune gene expression of pupae injected with fungal cell wall components.
DOI: https://doi.org/10.7554/eLife.32073.012

**Figure supplement 4.** Injection of fungal cell wall components increases abundance of chemical cues that are increased on unpacked pupae.
DOI: https://doi.org/10.7554/eLife.32073.013

*supplement 1*), this result indicates that the ants use one or more cuticular compounds to detect the infections.

Gas chromatography-mass spectrometry (GC–MS) analysis of the solvent wash confirmed that unpacked pupae have distinct chemical profiles compared to non-infected control pupae, whilst

cocooned (non-unpacked) pathogen-exposed pupae were intermediate (*Figure 2B*, *Figure 2—figure supplement 2*; perMANOVA: *F* = 1.49, df = 46, p=0.002; post-hoc perMANOVA comparisons: unpacked vs. control, p=0.003; unpacked vs. cocooned, p=0.79; cocooned vs. control, p=0.08). There were no novel compounds present on unpacked or cocooned pathogen-exposed pupae that were not also present on control pupae (*Table 1*; *Figure 2—figure supplement 2*), suggesting that these differences were not caused by odours emitted directly by the fungus, but were of pupal origin. By analysing the chemical profiles of each of the pathogen's separate developmental stages (infectious conidiospores, post-infection blastospores, and saprophytic mycelium) and performing a direct comparison of the fungal compounds to the pupal chemical profiles, we confirmed that there were no fungus-derived peaks in the pupal profiles (see Materials and methods for more information).

Most chemical messages in social insects are encoded by quantitative shifts of several compounds (*Leonhardt et al., 2016*). Correspondingly, we found that 8 out of the 24 CHCs identified (*Table 1*) had higher relative abundances on unpacked pupae compared to control pupae (*Figure 2C–F*, *Figure 2—figure supplement 2*; all Kruskal-Wallis [KW] test statistics and post-hoc comparisons in *Table 2*). Moreover, four of these CHCs were also present in relatively higher quantities on unpacked pupae compared to the non-unpacked cocooned pupae. Several specific CHCs are therefore probably accumulating on infected pupae over time, eventually reaching an amount that, relative to the

**Table 1.** Compound identification of cuticular hydrocarbons (CHCs) from *Lasius neglectus* pupae.
Table shows all 24 CHCs, with peak numbers listed in the order of their retention time, as in *Figure 3—figure supplement 2*. For comparability across gas chromatography–mass spectrometry systems, modified Kovats indices are included. Peaks that were significantly higher on unpacked pupae are highlighted in bold. In peaks 17 and 18, two compounds co-eluted.

| Peak # | Compound identification | Retention time (min) | Modified Kovats index |
|---|---|---|---|
| 1 | n-Heptacosane | 18.521 | 2699 |
| 2 | n-Octacosane | 18.883 | 2799 |
| 3 | n-Nonacosane | 19.253 | 2902 |
| 4 | 3-Methylnonacosane | 19.529 | 2974 |
| 5 | n-Triacontane | 19.624 | 2999 |
| 6 | n-Hentriacontane | 20.040 | 3100 |
| 7 | 3-Methylhentriacontane | 20.387 | 3175 |
| 8 | **Tritriacontadiene** | **20.764** | **3251** |
| 9 | **Tritriacontene** | **20.910** | **3279** |
| 10 | Tritriacontene | 20.958 | 3288 |
| 11 | n-Tritriacontane | 21.019 | 3300 |
| 12 | 13-Methyltritriacontane | 21.174 | 3326 |
| 13 | 3-Methyltritriacontene | 21.335 | 3353 |
| 14 | 3-Methyltritriacontane | 21.456 | 3373 |
| 15 | n-Tetratriacontane | 21.626 | 3402 |
| 16 | **Pentatriacontadiene** | **21.937** | **3447** |
| 17 | **Pentatriacontadiene + Pentatriacontene** | **22.134** | **3475** |
| 18 | n-Pentatriacontane + 13-Methylpentatriacontene | 22.306 | 3500 |
| 19 | 13,23-Dimethylpentatriacontane | 22.740 | 3554 |
| 20 | 11,25-Dimethylpentatriacontane | 22.752 | 3556 |
| 21 | 7,11,23-Trimethylpentatriacontane | 23.019 | 3589 |
| 22 | n-Hexatriacontane | 23.125 | 3602 |
| 23 | Unknown | 23.603 | 3652 |
| 24 | n-Heptatriacontane | 24.023 | 3697 |

DOI: https://doi.org/10.7554/eLife.32073.014

**Table 2.** Compounds contributing most to the differences between pupal cuticular hydrocarbon (CHC) profiles.
Table gives the overall effect of treatment per CHC, corrected for multiple testing, and the post-hoc comparisons, corrected at the level of each compound for multiple comparisons. CHCs significantly increased specifically on unpacked pupae shown in bold. All multiple comparison corrections performed using the Benjamini-Hochberg procedure ($\alpha = 0.05$).

| Peak # | Compound | KW $H$ (df = 2) | Corrected KW p value | Post-hoc comparison | Adjusted p value |
|---|---|---|---|---|---|
| 6 | n-Hentriacontane | 7.29 | 0.029 | Cocooned – Unpacked | 0.238 |
| | | | | Cocooned – Control | 0.309 |
| | | | | Unpacked – Control | 0.019 |
| 8 | **Tritriacontadiene** | 13.11 | 0.006 | **Cocooned – Unpacked** | **0.005** |
| | | | | **Cocooned – Control** | **0.830** |
| | | | | **Unpacked – Control** | **0.001** |
| 9 | **Tritriacontene** | 10.39 | 0.01 | **Cocooned – Unpacked** | **0.021** |
| | | | | **Cocooned – Control** | **0.745** |
| | | | | **Unpacked – Control** | **0.003** |
| 11 | Tritriacontane | 11.55 | 0.007 | Cocooned – Unpacked | 0.064 |
| | | | | Cocooned – Control | 0.245 |
| | | | | Unpacked – Control | 0.001 |
| 14 | 3-Methyltritriacontene | 7.63 | 0.028 | Cocooned – Unpacked | 0.428 |
| | | | | Cocooned – Control | 0.143 |
| | | | | Unpacked – Control | 0.021 |
| 16 | **Pentatriacontadiene** | 18.83 | 0.001 | **Cocooned – Unpacked** | **0.004** |
| | | | | **Cocooned – Control** | **0.152** |
| | | | | **Unpacked – Control** | **<0.001** |
| 17 | **Pentatriacontadiene +Pentatriacontene** | 12.09 | 0.007 | **Cocooned – Unpacked** | **0.039** |
| | | | | **Cocooned – Control** | **0.301** |
| | | | | **Unpacked – Control** | **0.001** |
| 18 | n-Pentatriacontane + 13-Methylpentatriacontene | 10.12 | 0.01 | Cocooned – Unpacked | 0.083 |
| | | | | Cocooned – Control | 0.312 |
| | | | | Unpacked – Control | 0.003 |

DOI: https://doi.org/10.7554/eLife.32073.015

other compounds, is sufficient to elicit destructive disinfection. This corresponds to current models of social insect behaviour, where the likelihood of a response depends on stimuli exceeding a certain threshold (*Theraulaz et al., 1998*; *Beshers and Fewell, 2001*).

To investigate the possibility that CHC changes on unpacked pupae are the result of an immune response developed by the host, we injected pupae with $\beta-1,3$-glucans – polysaccharides that are an integral component of fungal cell walls, including *Metarhizium* (*Wang and St Leger, 2006*). β-Glucans act as highly conserved major pathogen-associated molecular patterns (PAMPs) that are recognised by the immune system of invertebrates and can therefore be used to elicit an immune response in the absence of a pathogen (*Brown and Gordon, 2005*). We found that, within 2 days of injection, β-glucan , but not saline , caused an increase in the expression of immune genes, namely, an IMD signalling pathway regulator gene (*PGRP-SC2* [*Bischoff et al., 2006*]) and a gene encoding for a protein that recognises and binds to β-glucans to elicit an immune response (*β−1,3-GBP* [*Ma and Kanost, 2000*; *Gottar et al., 2006*]), whilst expression of the gene encoding for the melanisation cascade enzyme phenoloxidase (*proPO* [*Cerenius and Söderhäll, 2004*]) was unaffected (*Figure 2—figure supplement 3*). β-glucan injection also altered the chemical profiles of pupae in a similar way to *Metarhizium* infection (*Figure 2C–F*). Two of the four compounds we identified as a potential sickness cue on the unpacked pupae (Tritriacontadiene and Tritriacontene) were also increased in abundance within 2 days of injection with β-glucan, whilst there was no such increase in

control pupae (*Figure 2—figure supplement 4*). These data reveal that some of the changes in pupal chemical profile can be directly linked to a host reaction to an immune elicitor, similar to findings in honeybees (*Richard et al., 2012*; *Richard et al., 2008*), mice (*Arakawa et al., 2011*) and humans (*Shirasu and Touhara, 2011*).

## Destructive disinfection prevents pathogen replication

We next tested if destructive disinfection prevents pupal infections from replicating and becoming infectious. Pathogen-exposed pupae were kept with groups of ants (eight ants per pupae per group) until unpacking. They were then left with the ants for a further 1 or 5 days before being removed and incubated for fungal growth. We compared the number that subsequently sporulated to pathogen-exposed pupae kept without ants. Whilst 88% of pupae contracted infections, destructive disinfection significantly reduced the proportion of pupae that sporulated and hence became infectious (*Figure 3A*; GLM: LR $\chi^2$ = 40.47, df = 2, p<0.001; Tukey post-hoc comparisons: 1 vs. 5 d, p=0.04; all others, p<0.001). After only 1 day, the number of destructively disinfected pupae that sporulated decreased by 65%. With more time, the ants could reduce the number of pupae sporulating even further by 95%. Since the pupae were removed from the ants for fungal incubation, we can conclude that destructive disinfection permanently prevents pathogen replication. We repeated this experiment with a smaller number of ants (three ants per pupae per group) to investigate how group size influences the success of destructive disinfection. Smaller groups of ants were less efficient than larger ones: although they could still inhibit >90% of pupal infections within 5 days of unpacking, pupae tested for infection after 1 day still sporulated 70% of the time (*Figure 3—figure supplement*

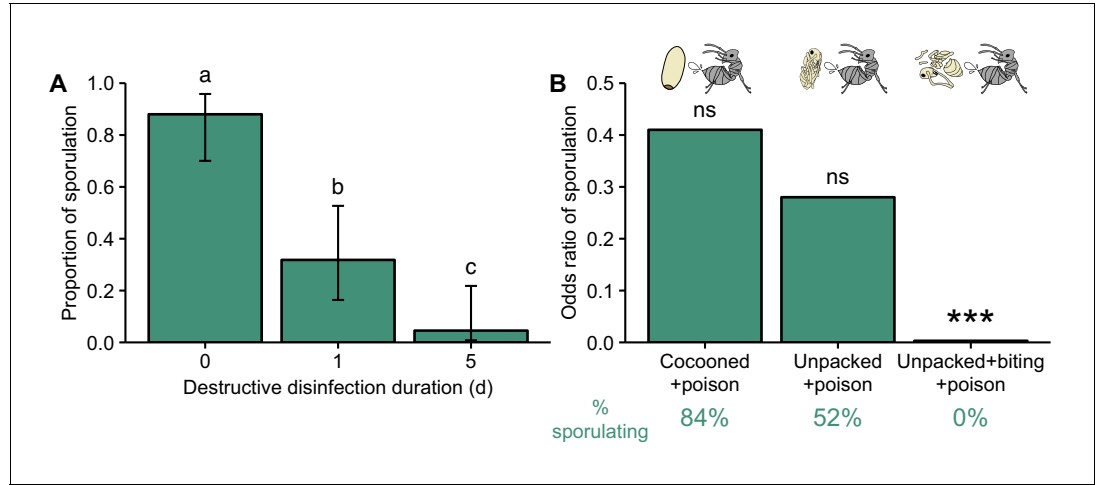

**Figure 3.** Destructive disinfection by ants prevents pathogen replication. (A) Destructive disinfection greatly reduced the probability of pupae sporulating compared to pupae that received no destructive disinfection (time point 0), and its effectiveness increased with the length of time ants could perform destructive disinfection (1 vs. 5 days; error bars show ± 95% CI; letters denote groups that differ significantly in Tukey post-hoc comparisons [p<0.05]). (B) The individual components of destructive disinfection (unpacking, biting and poison spraying) interacted to inhibit pathogen replication (% of pupae sporulating in each treatment shown under graph in green). The odds of sporulation for cocooned and unpacked pupae treated with poison were not significantly different to those of control pupae (cocooned pupae treated with water). But when unpacking, biting and poison spraying were combined, the odds of sporulation were significantly reduced (logistic regression; ns = non-significant deviation from control, ***=p<0.001; complete data set of full factorial experiment displayed in *Figure 3—figure supplement 3* and all statistics in *Table 3*).

DOI: https://doi.org/10.7554/eLife.32073.016

The following figure supplements are available for figure 3:

**Figure supplement 1.** Destructive disinfection of infected pupae in small groups of ants is less efficient.
DOI: https://doi.org/10.7554/eLife.32073.017
**Figure supplement 2.** Comparison of ant and synthetic poison spraying.
DOI: https://doi.org/10.7554/eLife.32073.018
**Figure supplement 3.** Destructive disinfection by ants prevents pathogen replication.
DOI: https://doi.org/10.7554/eLife.32073.019
**Figure supplement 4.** The pupal cocoon blocks the application of poison.
DOI: https://doi.org/10.7554/eLife.32073.020

*1*; GLM: LR $\chi^2$ = 35.23, p<0.001; Tukey post-hoc comparisons: 0 vs. 1 day, p=0.2; 0 vs. 5 days, p<0.001; 1 vs. 5 days, p=0.002). As the effectiveness of destructive disinfection increased with the amount of time the ants had, as well as with the number of ants present, we inferred that there must be a limiting factor affecting the inhibition the pathogen.

To study the underlying mechanisms of destructive disinfection, we performed its different components – unpacking, biting and poison spraying – in vitro to test for their relative importance and potential synergistic effects. We simulated unpacking by removing the cocoons of the pupae manually with fine forceps, and the cuticle damage caused by biting using dissection scissors. Previous work establishing the composition of *L. neglectus* poison (*Tragust et al., 2013a*) allowed us to create a synthetic version for use in this experiment (60% formic acid and 2% acetic acid, in water; applied at a dose equivalent to what ants apply during destructive disinfection; *Figure 3—figure supplement 2*), with water as a sham control. We then performed these 'behaviours' in different combinations in a full-factorial experiment. We found that all three behaviours must be performed in the correct order and interact to prevent pathogen replication (overview graph showing odds ratios of sporulation in *Figure 3B*, full data dataset displayed in *Figure 3—figure supplement 3*; GLM: overall LR $\chi^2$ = 79.9, df = 5, p<0.001; interaction between behaviours LR $\chi^2$ = 20.6, df = 2, p<0.001; all post-hoc comparisons in *Table 3*). As in sanitary care, the poison was the active antimicrobial compound that inhibited fungal growth (*Figure 3—figure supplement 3*, *Table 3*, *Tragust et al., 2013a*; *Graystock and Hughes, 2011*). However, for the poison to function the pupae had to be removed from their cocoons and their cuticles damaged. Firstly, this is because the cocoon itself is hydrophobic and thus prevents the aqueous poison from reaching the pupae inside (*Figure 3—figure supplement 4*). Secondly, as the infection is growing internally at the time of unpacking, the cuticle must be broken in order for the poison to enter the hemocoel of the pupae. This is achieved with the perforations created by the ants biting the pupal cuticle. It is possible that in the wild biting also helps to desiccate the pupae, since high levels of humidity are important for *Metarhizium* growth (*Doberski, 1981*). However, in our experiments, the relative humidity inside the petri dishes

**Table 3.** Tukey post-hoc comparisons between in vitro chemical treatments and pupa manipulations. Following a GLM showing a significant interaction between chemical treatment (water or synthetic poison) and pupae manipulation (cocooned, experimentally unpacked or experimentally unpacked and bitten), we performed Tukey post-hoc comparisons to determine the influence of each behavioural component. Comparisons to pupae that received complete destructive disinfection (unpacked + poison + biting) are shown in bold. All p values are corrected for multiple testing using the Benjamini-Hochberg procedure (α = 0.05).

| Post-hoc comparison | | Corrected p value |
|---|---|---|
| Cocooned + water | Cocooned + poison | 0.26 |
| Cocooned + water | Unpacked + water | 0.50 |
| Cocooned + water | Unpacked + poison | 0.05 |
| Cocooned + water | Unpacked + water + biting | 0.28 |
| **Cocooned + water** | **Unpacked + poison + biting** | **0.002** |
| Unpacked + water | Unpacked + poison | 0.02 |
| Unpacked + water | Cocooned + poison | 0.08 |
| Unpacked + water | Unpacked + water + biting | 0.61 |
| **Unpacked + water** | **Unpacked + poison + biting** | **0.001** |
| **Biting + water** | **Unpacked + poison + biting** | **0.001** |
| Biting + water | Cocooned + poison | 0.04 |
| Biting + water | Unpacked + poison | 0.01 |
| Cocooned + poison | Unpacked + poison | 0.37 |
| **Cocooned + poison** | **Unpacked + poison + biting** | **0.01** |
| **Unpacked + poison** | **Unpacked + poison + biting** | **0.02** |

DOI: https://doi.org/10.7554/eLife.32073.021

was always a stable 95%, so desiccation cannot have played a role in fungal inhibition. As the active antimicrobial component, we concluded that the poison is probably the limiting factor determining whether destructive disinfection is successful. Because the poison has a slow biosynthesis and each ant can only store a limited amount (*Tragust et al., 2013a*; *Hefetz and Blum, 1978*), it would explain why destructive disinfection was more likely to be successful the longer the ants had to treat the pupae, and as the number of ants increased (*Figure 3A*, *Figure 3—figure supplement 1*). By sharing the task of poison synthesis and application, the ants probably increase their chances of preventing the pathogen becoming infectious.

## Disruption of the pathogen lifecycle stops disease transmission

Finally, we investigated the impact of destructive disinfection on disease transmission within a social group. We created mini-nests comprising two chambers and a group of ants (five ants per group). Into one of the chambers we placed an infectious sporulating pupa – simulating a failure of the ants to detect and destroy the infection – or a pupa that had been destructively disinfected, and was thus non-infectious. The ants groomed, moved around and sprayed both types of corpses with poison. In the case of the sporulating pupae, all conidiospores were removed from the corpse by the ants. As in previous studies, sporulating corpses were highly virulent (*Hughes et al., 2002*; *Loreto and Hughes, 2016*) and caused lethal infections that became contagious after host death in 42% of ants (*Figure 4A*). However, there was no disease transmission from destructively disinfected pupae (*Figure 4A*; GLM: LR $\chi^2$ = 31.32, df = 1, p<0.001). We therefore concluded that by preventing the pathogen from completing its lifecycle destructive disinfection stops intra-colony disease transmission (*Figure 4B*).

## Discussion

In this study, we show that the superorganismal societies of ants have evolved an efficient mechanism to specifically target and eliminate infections that have established in colony members, before they become contagious. This is achieved through the detection of chemical cues emitted by infected pupae during the non-transmissible incubation period of the pathogen (*Figure 2*). The ants then engage in destructive disinfection, a multicomponent behaviour that utilises the ants' antimicrobial poison, in conjunction with cocoon removal and biting, to prevent pathogen replication within the body of the pupae (*Figure 1*, *Figure 3*). Ultimately, this prevented the pathogen from completing its lifecycle and infecting new hosts, thereby effectively reducing pathogen fitness to zero (*Figure 4*). These findings show that ants do not only avoid, groom and isolate pathogens (*Cremer et al., 2007*; *Cremer et al., 2017*) but can detect and eliminate infections developing inside the bodies of their nestmates, even before they have shown external disease symptoms.

Whilst the role of ant poison as a topical disinfectant by ants and other animals (i.e. 'anting' behaviour in birds) is well characterised (*Clayton et al., 2010*; *Verderane et al., 2007*; *Tragust, 2016*), its use as an internal disinfectant within the body of others during destructive disinfection is a novel and a rare example of the kill-component of social immunity (*Cremer and Sixt, 2009*). Eliminating infected kin to protect the rest of the group, observed in termites and honeybees as well (*Rothenbuhler, 1964*; *Chouvenc and Su, 2012*; *Spivak and Gilliam, 1998*), requires an unconditional level of altruism that is expected to be absent or at least rare in other forms of sociality (e.g. aggregations, non-superorganismal family groups and communal breeders [*Cremer et al., 2017*]), but has parallels to the immune system of the metazoan body (*Cremer and Sixt, 2009*). Both the immune system and social immunity have first lines of defences that reduce the risk of infection: pathogens that enter the body are met with mechanical and chemical defences, such as ciliated cells in the lung that move pathogens trapped in mucus out of the body (*Cremer and Sixt, 2009*), and in ants, sanitary care plays an analogous role (*Tragust et al., 2013a*; *Graystock and Hughes, 2011*). However, if a pathogen circumvents these first defences in the body and an infection occurs, the second line of defence is often a targeted elimination of the infected cells. This starts with immune cells detecting an infection and then transporting cell death-inducing and antimicrobial compounds into infected cells by creating pores in their membrane (*Walch et al., 2014*; *Kägi et al., 1994*; *Chowdhury and Lieberman, 2008*). Likewise, our experiments revealed that ants unpack infected pupae and make perforations in their cuticle, enabling the ants to spray their poison directly into the

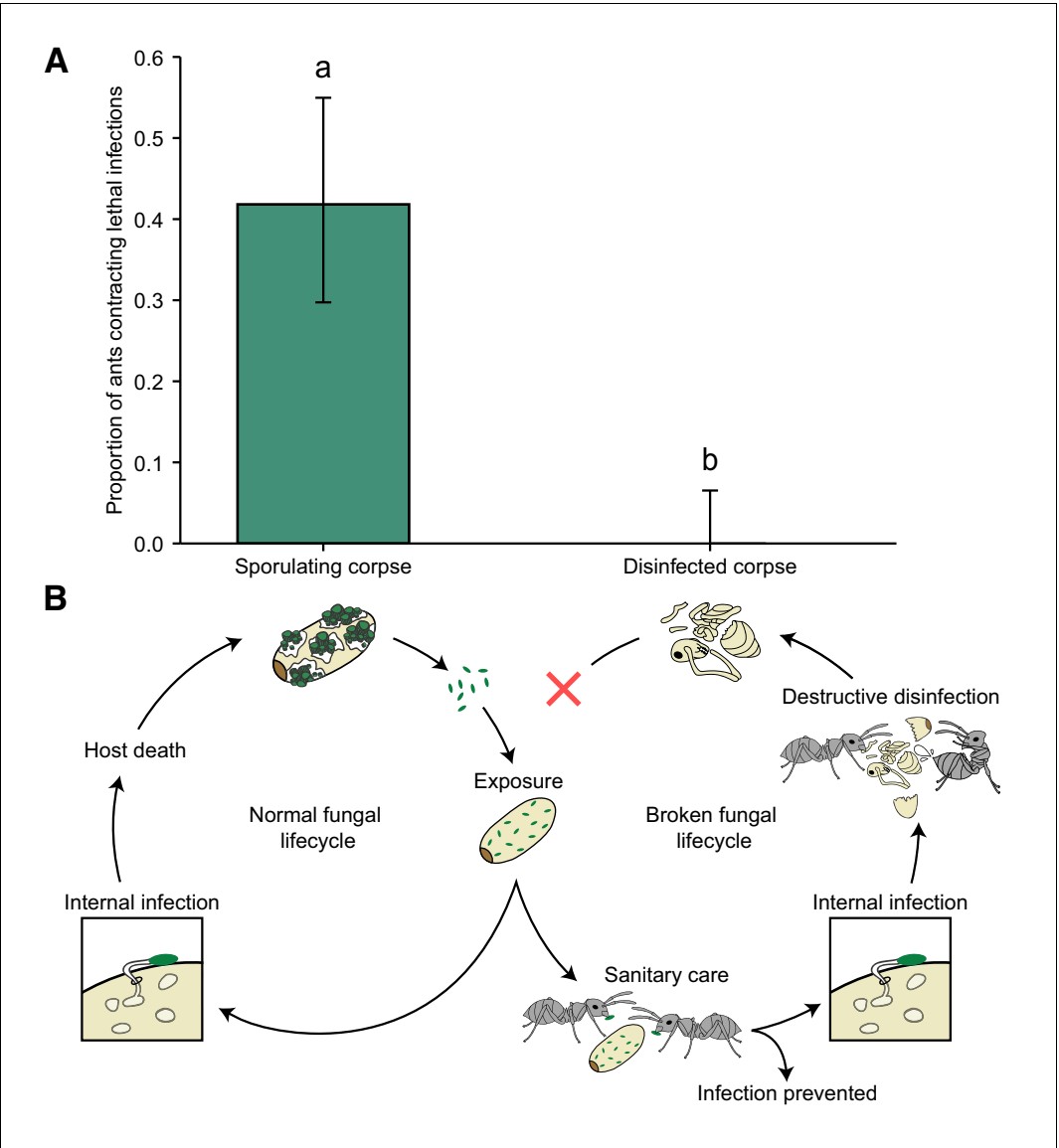

**Figure 4.** Destructive disinfection stops disease transmission. (**A**) Ants that interacted with sporulating pupae contracted lethal infections and died from fungal infection in 42% of the cases, whilst there was no disease transmission from destructively disinfected pupae (error bars show ± 95% CI; letters denote groups that differ significantly in a logistic regression [p<0.05]). (**B**) Overview of normal fungal lifecycle resulting in infectious, sporulating corpses (left) and a broken lifecycle due to the interference of the ants (right). When sanitary care fails to prevent infection in pathogen-exposed individuals, the ants switch to colony-level disease control, that is destructive disinfection to stop pathogen replication, resulting in non-infectious corpses.

DOI: https://doi.org/10.7554/eLife.32073.022

pupae's body; hence they display mechanisms analogous to immune cell elimination. In both cases, this second line of defence destroys the infected cell/insect, along with the infection, to prevent transmission (*Shore et al., 1976*). Since the loss of both somatic cells and individual insect workers can be tolerated with negligible effects on fitness (*Cremer and Sixt, 2009*), these convergent strategies have likely evolved at both the multicellular and superorganismal levels of biological organisation, as an effective way to clear infections and avoid any further damage to the body and colony, respectively.

Animals from a variety of taxa are known to identify sick conspecifics based on odour signals (*Bozza, 2015*; *Poirotte et al., 2017*; *Kiesecker et al., 1999*; *Anderson and Behringer, 2013*;

*Shirasu and Touhara, 2011*; *Swanson et al., 2009*), and although it has been hypothesised that ants should also use chemical cues to detect sick colony members, evidence has so far been lacking (*Ugelvig et al., 2010*; *Leclerc and Detrain, 2016*; *Bos et al., 2012*). To our knowledge, we have therefore observed ants using chemical information for the first time to rapidly and accurately target infected individuals. We found that the chemical compounds with increased abundances on infected pupae are distinct from those that induce the removal of corpses in ants (*Diez et al., 2013*; *Qiu et al., 2015*; *Wilson et al., 1958*), and, like in tapeworm-infected ants (*Trabalon et al., 2000*), are not pathogen-derived. This alteration of the hosts' chemical profile may arise during infection from the breakdown of hydrocarbons by *Metarhizium* penetration (*Lin et al., 2011*) or after infection due to an immune response affecting the synthesis of specific hydrocarbons (*Richard et al., 2012*; *Richard et al., 2008*). The latter is more likely because the ants only display destructive disinfection once the fungus is growing inside the pupae and injection of a fungal immune elicitor in the absence of live pathogen induced similar changes. The four CHCs specifically increased on unpacked pupae are all long-chained CHCs (carbon chain length $C_{33-35}$) with a low volatility, meaning that the ants have to be close to or touching the pupae to detect them (*Sharma et al., 2015*). As ants keep pupae in large piles, using low-volatility CHCs may be important so that the ants can accurately identify the infected pupae and do not mistakenly destroy healthy ones. Interestingly, two of the four CHCs that were increased on infected pupae, as well as on pupae that were injected with the fungal cell wall component, also had higher abundances on virus-infected honeybees (Tritriacontadiene [*Baracchi et al., 2012*]), or their brood when injected with a bacterial cell membrane component (Tritriacontene [*Richard et al., 2012*]). This raises the possibility that these hydrocarbons are evolutionarily conserved 'sickness cues' in Hymenopteran social insects. Such cues may have evolved into general sickness signals in social insects as they alert their kin to the presence of a developing infection that will harm the colony if it spreads (*Shakhar and Shakhar, 2015*). Relying on cues generated by an immune response to detect infections, over pathogen-specific cues, is likely more robust and general, similar to the expression of 'find-me/eat-me' signals by infected cells in vertebrate bodies (*Ravichandran, 2010*; *Grimsley and Ravichandran, 2003*), as well as the immune system responding to cell damage signals (danger signal hypothesis [*Matzinger, 2007*]). Such signals will be selected for in social insects because they can enhance colony fitness (and hence the indirect fitness of the sick individual) by preventing a systemic infection. Therefore, altruistic displays of sickness can evolve in superorganisms, even if this results in the destruction of the individual that expresses them (*Cremer et al., 2017*).

Our experiments show that destructive disinfection was highly effective and prevented 95% of infections becoming transmissible. Destructive disinfection will thus keep the average number of secondary infections caused by an initial infection low and the disease will die out within the colony (*Schmid-Hempel, 2017*). This may explain why infections of *Metarhizium* and other generalist entomopathogenic fungi like *Beauveria*, though common in the field (*Reber et al., 2012*; *Hughes et al., 2004a*; *Cremer et al., 2008*; *Keller et al., 2003*), do not seem to cause colony-wide epidemics in ants (*Cremer et al., 2017*), but are more numerous in solitary species that lack social adaptations to resist disease (*Roberts and St Leger, 2004*; *Shimazu, 1989*; *Lomer et al., 1997*). Behaviours like destructive disinfection that are able to reduce pathogen fitness to zero could have selected for host manipulation in fungi that specialise on infecting ants, for example *Ophiocordyceps* and *Pandora* (*Hughes et al., 2016*; *Loreto et al., 2014*; *Małagocka et al., 2017*). These fungi manipulate their ant hosts into leaving the nest before they become infectious and our study supports previous suggestions that this may be to avoid social immunity defences, like destructive disinfection, which would prevent them completing their lifecycle inside the nest (*Loreto et al., 2014*). In contrast to specialists, generalist pathogens like *Metarhizium* infect a broad range of solitary and social hosts, making it less likely that they evolve strategies to escape social immunity defences (*Cremer et al., 2017*; *Agosta et al., 2010*). We have also observed destructive disinfection in an unrelated supercolonial population of *L. neglectus* (*Ugelvig et al., 2008*) and another non-supercolonial/non-invasive *Lasius* species, *L. niger* (see Materials and methods for more information), suggesting that it may be a common behaviour that has evolved in *Lasius*, and possibly other ant genera, in response to the constant selection pressure applied by generalist pathogenic fungi (*Cremer et al., 2017*). Future work that investigates how social immunity disrupts typical host-pathogen dynamics will shed light on the co-evolution of pathogens and their social hosts (*Schmid-Hempel, 2017*).

Destructive disinfection has probably evolved in ants because the removal of corpses from the colony alone does not guarantee that disease transmission is prevented (*Loreto et al., 2014*). Ants are usually sedentary, building nests that remain in the same location until the colony dies (*Boomsma et al., 2005*; *Schmid-Hempel, 1998*). Additionally, they are highly territorial and forage mostly in the area around their nests, meaning that if they do not clear it of dead and potentially infectious nestmates, they are likely to be reencountered (*Boomsma et al., 2005*; *Cremer et al., 2017*). Ants tend therefore to place corpses onto specific midden (trash) sites that are located inside or outside near the nest, but these sites are still regularly visited by midden workers (*Verza et al., 2017*; *Hart and Ratnieks, 2002*; *Farji-Brener et al., 2016*). Consequently, although middens likely reduce a colony's exposure to corpses, they still represent a potential route for disease transmission back into the colony; hence the need to destroy infected corpses rather than simply taking them out of the nest. This is in contrast to honeybees, where corpses are dumped randomly outside of the hive (*Wilson-Rich et al., 2009*). This behaviour is sufficient to prevent disease transmission because honeybees are unlikely to reencounter corpses whilst foraging on the wing (*Spivak and Reuter, 2001*). Termites on the other hand cannibalise their dead (*Chouvenc and Su, 2012*; *Rosengaus and Traniello, 2001*). Cannibalism is effective because the termite gut and/or its microbiome neutralises ingested pathogens (*Chouvenc et al., 2009*; *Rosengaus et al., 2014*; *Rosengaus et al., 1998*) and has likely evolved because dead nestmates are a source of valuable nitrogen in their cellulose-base diet (*Rosengaus et al., 2011*). The same selective pressure has driven this suite of independently evolved innovations – the need to eliminate or remove infected individuals early in the infectious cycle – with the ants expressing a particularly complex behavioural repertoire. This seems to be a general principle in disease defence, as cells are also rapidly detected and destroyed shortly after infection to prevent pathogen spread in multicellular organisms (*Cremer and Sixt, 2009*).

Understanding how natural selection can result in similar traits at different levels of biological organisation and in organisms with different life histories is a central question in evolutionary biology (*Bourke, 2011*). Studying the similarities and differences between organismal immunity and social immunity could therefore potentially lead to new insights about how disease defences evolve (*Cremer and Sixt, 2009*; *Kennedy et al., 2017*). For example, in most social animals, where related-ness is low or altruism is only conditionally expressed (e.g. by young that eventually disperse to reproduce themselves), disease defences tend to rely on self-protective infection avoidance (*Curtis, 2014*), mutually expressed sanitary care (*Nunn and Altizer, 2006*) and herd immunity only (*John and Samuel, 2000*). In contrast, the results of our study suggest that equivalent selection pressures during the major evolutionary transitions from unicellularity to multicellularity in the metazoans, and from sociality to superorganismality in the social insects, have resulted in convergent defences that protect multicellular organisms and superorganismal insect societies from systemic disease spread, by ensuring the survival of the whole over its parts (*Cremer and Sixt, 2009*).

## Materials and methods

### Ant host

We studied the invasive garden ant *Lasius neglectus* (*Cremer et al., 2008*) that forms large, under-ground nests in the soil. Populations of this species lack territorial structuring and instead consist of interconnected nests, forming a single supercolony between which there is a constant exchanging of individual ants (*Cremer et al., 2008*). We sampled more than one hundred queens, many thousands of workers and hundreds of brood items from a 320 m$^2$ area of the supercolony in Seva, Spain (41°48'32.4"N 2°15'43.9"E), and reared them as stock colonies in the laboratory. All experiments were conducted in plastered petri dishes (Ø=33, 55 or 90 mm) with 10% sucrose solution provided *ad libitum* and environmental conditions were controlled throughout (23°C; 70% RH; 14/10 hr light/dark cycles). In addition, we directly measured the humidity of plastered petri dishes without ants for 2 weeks, by embedding a digital relative humidity sensor (Sensirion, Switzerland) into the lids of the dishes, finding that the relative humidity inside was always a stable 95%. The animal use protocol was performed in accordance with the IST Austria Ethics Committee guidelines. At present, the committee does not provide a specific approval numbers for invertebrate animal research. Animals used in this study, *Lasius neglectus*, do not belong to regulated or protected species.

## Fungal pathogen

As a model pathogen, we used an obligately killing pathogen of *Lasius* ants, *Metarhizium brunneum* (CDP unpublished data; strain MA275, KVL 03–143). Entomopathogenic *Metarhizium* fungi occur at high densities in the soil (up to 5000 conidiospores/g soil [*Keller et al., 2003*]) and on sporulating cadavers (up to 12 million conidiospores/cadaver [*Hughes et al., 2002*, *2004b*]) and are responsible for natural infections of ants in field populations (*Reber et al., 2012*; *Hughes et al., 2004a*). *Metarhizium* and other similar generalist fungi are therefore expected to have applied a persistent and pervasive selection pressure on ants over the course of their evolutionary history (*Boomsma et al., 2005*; *Cremer et al., 2017*). Multiple aliquots of conidiospore suspensions were kept in long-term storage at – 80°C. Prior to each experiment, the conidiospores were grown on sabaroud dextrose agar at 23°C until sporulation and harvested by suspending them in 0.05% sterile Triton X-100 (Sigma Aldrich, Austria). The germination rate of conidiospore suspensions was determined before the start of each experiment and was >90% in all cases. In addition to fungal conidiospores, we also cultured blastospores and mycelia to obtain the chemical profiles (see below) of all stages of the fungi's lifecycle (*Deacon, 2006*). Blastospores were cultured by adding 50 µl of conidiospore suspension ($10^9$/ml) to 50 ml of Adámek liquid media (with Streptomycin sulphate [0.005 g/l] and Chloramphenicol [0.025 g/l] added to inhibit bacterial growth) in a 300 ml Erlenmeyer flask, which was then incubated (72 hr, 200 rpm, 23°C) (*Adámek, 1965*; *Kleespies and Zimmermann, 1992*). After incubation, the liquid (which contains the blastospores) was pre-filtered using a flame-sterilised mesh and sieve and the liquid vacuum-filtered (40 µm mesh; Millipore Steriflip [Merck, Germany]). The resulting blastospore suspension was then washed (5 min, 3000 g, 23°C) three times in PBS. To produce mycelia, we added 50 µl of $10^6$/ml conidiospore suspension to 100 ml of YPD liquid broth (Yeast extract Peptone Dextrose with Streptomycin sulphate [0.005 g/l] and Chloramphenicol [0.025 g/l]) in a 300 ml Erlenmeyer flask. We incubated the flask (5 days, 180 rpm, 27°C) and vacuum-filtered (40 µm mesh; Millipore Steriflip) the resulting fungal mass to remove the liquid broth. We then washed the mycelial mass three times in autoclaved distilled water.

## Pupal pathogen exposure

Conidiospores were applied in a suspension of 0.05% autoclaved Triton-X 100 at a concentration of $10^6$ conidiospores/ml in all experiments, unless otherwise stated. Throughout the study, we used cocooned worker pupae of approximately the same age, which was determined by assessing the melanisation of the eyes and cuticle. Single pupae were exposed by gently rolling them in 1 µl of the conidiospore suspension using sterile soft forceps. Pupae were then allowed to air dry for 5–10 min before being used in experiments. This exposure procedure resulted in pupae receiving ~1800 conidiospores, of which 5% (~95 conidiospore) passed through the cocoon and came into contact with the pupa inside (*Figure 1—figure supplement 1*). In all experiments, pupae were allocated to treatment groups haphazardly.

## Statistical analysis

Statistical analyses were carried out in R version 3.3.2 (*Core Team, 2012*) and all tests were two-tailed. All General(ised) linear and mixed models were compared to null (intercept only) and reduced models (for those with multiple predictors) using Likelihood Ratio (LR) tests to assess the significance of predictors (*Bolker et al., 2009*). We controlled for the number of statistical tests performed per experiment to protect against a false discovery rate using the Benjamini-Hochberg procedure ($\alpha$ = 0.05). Moreover, all post-hoc analyses were corrected for multiple testing using the Benjamini-Hochberg procedure ($\alpha$=0.05) (*Benjamini and Hochberg, 1995*; *García, 2004*). We checked the necessary assumptions of all tests that is by viewing histograms of data, plotting the distribution of model residuals, checking for non-proportional hazards, testing for unequal variances, testing for the presence of multicollinearity, testing for over-dispersion, and assessing models for instability and influential observations. For mixed effects modelling, we used the packages 'lme4' to fit models (*Bates et al., 2014*), 'influence.ME' to test assumptions (*Nieuwenhuis and Pelzer, 2012*), and, for LMERs, 'lmerTest' to obtain p values (*Kuznetsova et al., 2015*). All logistic regressions were performed using either generalised linear models (GLMs) or generalised linear mixed models (GLMMs), which had binomial error terms and logit-link function. The Cox proportional hazards regression was carried out using the 'coxphf' package with post-hoc comparisons achieved by re-levelling the

model and correcting the resulting p values (*Ploner and Heinze, 2015*). For Kruskal-Wallis (KW) tests and subsequent post-hoc comparisons we used the 'agricolae' package, which implements the Conover-Iman test for multiple comparisons using rank sums (*de Mendiburu, 2016*). For the perMANOVA, we used the package 'vegan' and performed pairwise perMANOVAs for post-hoc comparisons (*Oksanen et al., 2016*). All other post-hoc comparisons were performed using the 'multcomp' package (*Bretz et al., 2011*). All graphs were made using the 'ggplot2' package (*Wickham, 2009*). Preliminary studies were performed for all major experiments to determine sample size. No data outliers were detected or removed and all replicate information represents biological replicates. Individual descriptions of statistical analyses are given for all experiments below.

## Unpacking behaviour

To study how ants respond to infections, we exposed pupae to a low ($10^4$/ml), medium ($10^6$/ml) or high ($10^9$/ml) dose of conidiospores or autoclaved Triton X as a sham control (sham control, $n$ = 24; all other treatments, $n$ = 25). The pupae were then placed into individual petri dishes with two ants and inspected hourly for 10 hr/days for 10 days. When the ants unpacked a pupa, it was removed and surface-sterilised to ensure that any fungal outgrowth was the result of internal infections and not residual conidiospores on the cuticle. To surface-sterilise, pupae were dipped in 70% ethanol, washed in autoclaved distilled water and submerged in 0.05% sodium hypochlorite for 1 min, before being washed three times in autoclaved distilled water (*Lacey and Brooks, 1997*). After sterilisation, we transferred the pupae to a petri dish lined with damp filter paper at 23°C and monitored them for 2 weeks for *Metarhizium* sporulation to confirm the presence of an internal infection (low dose, $n$ = 8; medium dose, $n$ = 18; high, $n$ = 21). In addition, any cocooned pupae that were not unpacked after 10 d were removed from the ants, surface sterilised and observed for sporulation, as above (low dose, $n$ = 11; medium dose, $n$ = 4; high, $n$ = 4). Some pupae (control = 16, low dose = 6, medium dose = 3) successfully emerged from the cocoon as adult ants and were thus treated as non-unpacked in analyses. We analysed the effect of treatment on unpacking using a Cox proportional hazards model with Firth's penalised likelihood, which offers a solution to the monotone likelihood caused by the complete absence of unpacking in the sham control treatment. We followed up this analysis with post hoc comparisons (model factor re-levelling) to test unpacking rates between treatments (*Figure 1B*). We compared the number of unpacked and cocooned pupae sporulating using a logistic regression, which included pupa type (cocooned, unpacked), conidiospore dose (low, medium, high) and their interaction as main effects. The interaction was non-significant (GLM: LR $\chi^2$ = 5.0, df = 2, p=0.084); hence, it was removed to gain better estimates of the remaining predictors.

## Images and scanning electron micrographs (SEMs) of destructive disinfection

Photographs of destructive disinfection were captured (Nikon D3200 [Nikon, Japan]) and aesthetically edited (Adobe Photoshop [Adobe Systems, San Jose, California]) to demonstrate the different behaviours (*Figure 1A*). They were not used in any form of data acquisition. We also made representative SEMs of a pupa directly after unpacking and one after destructive disinfection (24 hr after unpacking; *Figure 1F–G*). As the pupae were frozen at − 80°C until the SEMs were made, we also examined non-frozen pupae taken directly from the stock colony and confirmed that freezing itself does not cause damage to the pupa (not shown).

## Conidiospore load on unpacked pupae

We determined the number of conidiospores on unpacked pupae ($n$ = 7) and their removed cocoons ($n$ = 7) by placing them into separate vials containing 100 µl autoclaved 0.05% Trixton-X 100. The vials were then shaken for 10 m at 600 RPM (Vortex Genie 2 [Scientific Industries, Bohemia, New York]) and the resulting supernatant was plated onto selective medium agar. We counted the number of *Metarhizium* colony-forming units (CFUs) that subsequently grew on the plates after 7 d. As a control, we performed the same experiment on pupae directly after pathogen-exposure. We experimentally unpacked the pupae using sterile (ethanol wiped) forceps so that we could examine the number of CFUs present on the pupae ($n$ = 16) and cocoon separately ($n$ = 16). We analysed the

number of CFUs on pupae and cocoons using Mann-Whitney *U* tests (*Figure 1—figure supplement 1*).

## Comparison of sanitary care and destructive disinfection behaviours

To observe how the behavioural repertoire of the ants changes between sanitary care and destructive disinfection, we filmed three individually colour-marked ants tending a single pathogen-exposed pupa with a USB microscope camera (Di-Li 970-O [Di-Li, Germany]). To characterise the sanitary care behaviours of the ants, we analysed the first 24 hr of the videos following the introduction of the pupa. To study destructive disinfection behaviours, we analysed the 24 hr period that immediately followed unpacking. Videos were analysed using the behavioural-logging software JWatcher (*Blumstein and Daniel, 2007*). For each ant (*n* = 15), we recorded the duration of its grooming bouts, the frequency of poison application and the frequency of biting. Grooming duration was analysed using a LMER, having first log-transformed the data to fulfil the assumption of normality (*Figure 1C*). The frequency of poison spraying and biting (*Figure 1D–E*) were analysed using separate GLMMs with Poisson error terms for count data and logit-link function. We included an observation-level random intercept effect to account for over-dispersion in the poison spraying and biting data (*Harrison, 2014*). In all three models, we included petri dish identity as a random intercept effect because ants from the same dish are non-independent. Additionally, a random intercept effect was included for each ant as we observed the same individuals twice (before and after unpacking).

## Comparison of pupal mortality after unpacking and destructive disinfection

We established a protocol to determine whether pupae were dead or alive because it is not generally obvious when death has occurred. To ensure that we examined pupae as soon as possible after unpacking, we checked pathogen-exposed pupae housed with ants every 45 min for 15 hr/d. When unpacking occurred, we either removed the pupa immediately (*n* = 33) or left it with the ants for a further 24 hr so that they could perform destructive disinfection (*n* = 44). To check the numbers of dead and alive pupae at the time point of unpacking and after destructive disinfection, we secured the pupae to glass slides using double-sided tape. The pupae were then gently prodded with a glass capillary whilst being examined under a bifocal microscope (10 x magnification; Leica DM 1000 [Leica Biosystems, Germany]). If pupae were alive, this resulted in contractions of their dorsal aorta (*Broome et al., 1976*), which is visible through the cuticle of the abdomen. If they were dead, no contractions occurred. Each examination lasted a maximum of 5 min. To confirm that this approach was sensitive, we examined experimentally unpacked pupae taken straight from a stock colony (*n* = 10). In all cases, these pupae were alive. They were then frozen at – 80°C for 1 day and examined again after defrosting, when they were all found to be dead. We compared the number of dead pupae at the time point of unpacking to the number that were dead after destructive disinfection using a logistic regression (*Figure 1—figure supplement 2*). We included the day of unpacking as a covariate to test if pupae unpacked sooner or later were more or less likely to have already died.

## Estimation of poison load on pupae after destructive disinfection

As *L. neglectus* poison has a very high acidity (*Tragust et al., 2013a*), we could measure the pH of pupae to determine if ants apply higher amounts poison to pupae during destructive disinfection. We kept a pair of pathogen-exposed or sham control pupae with two ants. When one of the pathogen-exposed pupae in a pair was unpacked, we let the ants perform destructive disinfection for 24 hr (*n* = 25). In the control, we experimentally unpacked one pupa in a pair and placed it back with the ants for 24 hr (*n* = 17). After 24 hr, we removed the unpacked pupae in both treatments along with their discarded cocoons. At the same time, the second, still cocooned pupae in each pair was removed and experimentally unpacked so that pH measurements were consistent across pupal groups (pathogen exposed, *n* = 9; control, *n* = 16). All pupae and their cocoons were placed into individual vials containing 20 µl of autoclaved distilled water and a sterile glass pestle was used to crush each pupa and cocoon for 60 s. The pH of the resulting pupa/cocoon slurry was measured using a pH electrode meter (INLAB ULTRA-MICRO, SevenGo PRO pH SG8 pH-meter [Mettler-Toledo, Columbus, Ohio]). This gave us an indication of how much poison the ants had applied to

each type of pupa (*Figure 1—figure supplement 3*). We used a LMER with Tukey post-hoc comparisons to compare the pH measurements of the pupae. Pupa treatment (pathogen-exposed or control), type (cocooned or unpacked) and their interaction were included as main effects. Petri dish was included as a random intercept effect as pairs of pupae from the same dish are non-independent. As we used a portion of this dataset in *Figure 3—figure supplement 2*, we corrected the overall model p value for multiple testing.

## Chemical bioassay

We determined whether ants detect infected pupae through potential changes in the pupae's cuticular chemical profile. We established internal infections in pupae by exposing them to the pathogen and leaving them for 3 days in isolation. In pilot studies, approx. 50% of these pupae were then unpacked within 4 hr of being introduced to ants. After 3 days, pupae were washed for 2.5 min in 300 µl of either pentane solvent to reduce the abundance of all CHCs present on the pupae (*n* = 28), or in autoclaved water as a handling control (*n* = 28). After washing, pupae were allowed to air dry on sterile filter paper. Additionally, non-washed pupae were used as a positive control (*n* = 30). Pupae were placed individually with a pair of ants in petri dishes and observed for unpacking for 4 hr. We used GC–MS (see below for methodology) to confirm that washing was effective at removing cuticular compounds, by comparing the total amount of chemicals present on pupae washed in pentane to non- and water-washed pupae (*n* = 8 per treatment; *Figure 2—figure supplement 1*). The number of pupae unpacked between the different treatments was analysed using a logistic regression (*Figure 2A*). As several researchers helped to wash the pupae, we included a random intercept for each person to control for any potential handling effects. Additionally, the experiment was run in two blocks on separate days, so we included a random intercept for each block to generalise beyond any potential differences between runs. The total peak area from the GC–MS analysis was compared between treatments using a KW test with post-hoc comparisons.

## Chemical analysis of pupal hydrocarbon patterns

To confirm that infected pupae had chemical profiles that are different from pathogen-exposed cocooned and control pupae, we exposed pupae to the pathogen or a sham control. Pupae were then isolated for 3 days to establish infections in the pathogen-exposed treatment (as above). Following isolation, pupae were individually placed with ants and observed for unpacking for 4 hr. Unpacked pupae were immediately frozen at – 80°C with the removed cocoons (*n* = 13) and we also froze cocooned pathogen-exposed pupa that had not yet been unpacked (*n* = 10). Furthermore, we froze a pair of control pupae, of which one was cocooned (*n* = 12), whilst the other was first experimentally unpacked (to test if the cocoon affects cuticular compound extraction; *n* = 12). Cuticular chemicals were extracted from individual pupae and their cocoons in glass vials (1.8 ml [Supelco, Germany]) containing 100 µl n-pentane solvent for 5 min under gentle agitation. The vials were then centrifuged at 3000 rpm for 1 min to spin down any fungal conidiospores that might be remaining, and 80 µl of the supernatant was transferred to fresh vials with 200 µl glass inserts and sealed with Teflon faced silicon septa (both Supelco). The pentane solvent contained four internal standards relevant for our range of hydrocarbons ($C_{27}$ – $C_{37}$); n-Tetracosane, n-Triacontane, n-Dotriacontane and n-Hexatriacontane (Sigma Aldrich) at 0.5 µg/ml concentration, all fully deuterated to enable spectral traceability and separation of internal standards from ant-derived substances. We ran extracts from the different groups in a randomised manner, intermingled with blank runs containing only pentane, and negative controls containing the pentane plus internal standards (to exclude contaminants emerging for example from column bleeding), on the day of extraction, using GC–MS (GC7890 coupled to MS5975C [Agilent Technologies, Santa Clara, California]).

A liner with one restriction ring filled with borosilicate wool (Joint Analytical Systems, Germany) was installed in the programmed temperature vaporisation (PTV) injection port of the GC, which was pre-cooled to −20°C and set to solvent vent mode. 50 µl of the sample extractions were injected automatically into the PTV port at 40 µl/s using an autosampler (CTC Analytics, PAL COMBI-xt, , CHRONOS 4.2 software [Axel Semrau, Germany]) equipped with a 100 µl syringe (Hamilton [Sigma-Aldrich]). Immediately after injection, the PTV port was ramped to 300°C at 450 °C/min, and the sample transferred to the column (DB-5ms; 30 m × 0.25 mm, 0.25 µm film thickness) at a flow of 1 ml/min. The oven temperature program was held at 35°C for 4.5 min, then ramped to 325°C at 20°C/

min, and held at this temperature for 11 min. Helium was used as the carrier gas at a constant flow rate of 3 ml/min. For all samples, the MS transfer line was set to 325°C, and the MS operated in electron ionisation mode (70 eV; ion source 230°C; quadrupole 150°C, mass scan range 35–600 amu, with a detection threshold of 150). Data acquisition was carried out using MassHunter Workstation, Data Acquisition software B.07.01 (Agilent Technologies).

Analytes were detected by applying deconvolution algorithms to the total ion chromatograms of the samples (MassHunter Workstation, Qualitative Analysis B.07.00 [Agilent Technologies). Compound identification (*Table 1*) was performed via manual interpretation using retention indices and spectral information, and the comparison of mass spectra to the Wiley 9<sup>th</sup> edition/NIST 11 combined mass spectral database (National Institute of Standards and Technologies). As the molecular ion was not detectable for all analytes based on electronic ionisation, we in addition performed chemical ionisation on pools of 20 pupae in 100 μl n-pentane solvent with 0.5 μg/ml internal standards. The higher extract concentration was needed to counteract the loss in ionisation efficiency in chemical ionisation mode. A specialised chemical ionisation source with methane as the reagent gas was used with the MS, while the chromatographic method was the same as in electronic ionisation mode. Use of external standards ($C_7$-$C_{40}$ saturated alkane mixture [Sigma Aldrich]) enabled traceability of all peaks, and thus comparison to runs of single pupae extracts made in electronic ionisation mode. Modified Kovats retention indices for the peaks in question were calculated based on those standards. To further aid identification, we separated the substances based on polarity using solid phase extraction fractionation. For this purpose, pools of 20 pupae were extracted in 500 μl n-pentane containing 0.2 μg/ml internal standard, and separated on unmodified silica cartridges (Chromabond SiOH, 1 ml, 100 mg) based on polarity. Prior to use, the cartridges were conditioned with 1 ml dichloromethane followed by 1 ml n-pentane. The entire extraction volume was loaded onto the silica and the eluent (fraction 1, highly apolar phase) collected. A wash with 1 ml pure n-pentane was added to fraction 1. Fraction 2 contained all substances washed off the silica with 1 ml 25% dichloromethane in n-pentane, and finally a pure wash with 1 ml dichloromethane eluted all remaining substances (fraction 3). The polarity thus increased from fraction 1 through 3, but no polar substances were found. All fractions were dried under a gentle nitrogen stream and re-suspended in 70 μl n-pentane followed by vigorous vortexing for 45 s. GC–MS analysis of all fractions was performed in electronic ionisation mode under the same chromatographic conditions as before.

To quantify the relative abundances of all compounds found on each pupa, analyte-characteristic quantifier and qualifier ions were used to establish a method enabling automatised quantification of their integrated peak area relative to the peak area of the closest internal standard. For each analyte, the relative peak area was normalised, that is divided by the total sum of all relative peak areas of one pupa, to standardise all pupa samples. Only analytes, which normalised peak area contributed more than 0.05% of the total peak area, were included in the statistical analysis. We compared the chemical profiles of the pupae using a perMANOVA analysis of the Mahalanobis dissimilarities between pupae, with post hoc perMANOVA comparisons. Since there was no difference between cocooned and unpacked control pupae we combined them into a single control group for the final analysis (perMANOVA: $F$ = 1.09, df = 23, p=0.1). We also performed a discriminant analysis of principle components (*Figure 2B*) to characterise the differences between the pupal treatments (*De Moraes et al., 2014*; *Jombart et al., 2010*). To identify the compounds that differ between treatments, we performed a conditional random forest classification ($n$ trees = 500, $n$ variables per split = 4) (*De Moraes et al., 2014*; *Strobl et al., 2009a*; *Strobl et al., 2009b*). Random forest identified nine compounds that were important in classifying the treatment group, of which eight were significant when analysed using separate KW tests (results for significant compounds in *Table 2*). We followed up the KW tests with individual post hoc comparisons for each significant compound (*Figure 2C–F*, post-hoc comparisons in *Table 2*).

## Comparison of fungal and pupal chemical profiles

One millilitre aliquots of conidiospore ($10^9$/ml in 0.05% TX) and blastospore ($4 \times 10^6$/ml in PBS) suspensions and approx. 500 mg of mycelia (in 500 μl of autoclaved distilled water) were washed three times by briefly vortexing and centrifuging the samples (5 min, 5500 g), discarding the supernatants, and replacing with 1 ml of autoclaved distilled water for the first two washes, and 500 μl for the last wash. One hundred and fifty microliters of the conidiospore and blastospore suspensions and 155 μg of hyphae ($n$ = 3 for each fungal stage) were transferred into 1.5 ml glass vials (La-Pha-

Pack, Germany). All samples were centrifuged (2 min, 3000 g) and dried under a nitrogen stream for 2 hr. Once samples were dry, 200 µl n-pentane containing internal standards (as above) was added to the samples, which were vortexed for 2 min. Samples were centrifuged (5 min, 5000 g) and the supernatants transferred into 200 µl glass vials with inserts, and closed with aluminium crimper caps that had a silicone septum (both La-Pha-Pack). Fifty microliters of the samples were injected into a pre-cooled PTV inlet at – 20°C and GC–MS analysis carried out following the above protocol.

We determined if there was any overlap between the chemical profiles of pupae and the fungus by comparing the results of the fungal GC–MS analysis to the results of the pupal GC–MS. To that end, all fungal chromatograms were automatically de-convoluted and the mass-spectra of the compound peaks compared to a mass-spectral database, composed of the substances found on the pupae (Agilent Technologies MassHunter Qualitative Analysis, B.07.00, 2014). Twenty-seven compounds scored above 70 points, with 82.21 being the highest score. To determine if these peaks were identical to the peaks of the pupal samples, we calculated their Kovats retention time indices (RI) and compared them to that of the pupal substances. This analysis revealed that none of the fungal compound RIs were overlapping with the pupal compounds, hence confirming that the identified pupal substances are not of fungal origin.

## Immune stimulation of pupae using β-glucans

We injected β−1,3-glucans to test whether the changes in the chemical profile of infected pupae may be caused by an immune stimulation (*Vilcinskas and Wedde, 1997*; *Unestam and Söderhäll, 1977*; *Gunnarsson, 1988*). Soluble β−1,3-glucans were acquired by suspending 5 mg of Zymosan-A (*Saccharomyces cerevisiae* cell wall fragments [Sigma-Aldrich]) in 1 ml of sterile physiological ant saline (as described in [*Aubert and Richard, 2008*]). The Zymosan suspension was vortexed for 1 hr at 3200 rpm before being centrifuged at 10000 rcf for 5 min. The supernatant that contains the soluble β-glucans (*Vilcinskas and Wedde, 1997*) was then removed and stored at 4°C until use. As a control we used sterile ant physiological saline (*Aubert and Richard, 2008*). Pupae were artificially unpacked from their cocoons (as above) and placed gently into a sponge harness. Using fine glass capillaries (with spike to aid injection; inner diameter = 25 µm [BioMedical Instruments, Germany]), a microinjector (parameters: pi = 120 hPa, ti = 0.3 s, pc = 20 hPa [FemtoJet, Eppendorf, Germany]) and a micromanipulator (Luigs and Neumann, Germany), we injected 46 nl of the β-glucan solution or ant physiological saline through the pupae's first tergite, into their haemocoel. We cleaned the capillaries between injections using 96% ethanol. Half of the pupae were frozen at – 80°C immediately after injection whilst the remainder were kept alone in individual plaster dishes for a further 48 hr, before then also being frozen. Frozen pupae were then used for molecular and chemical analyses (below).

## Immune gene expression of pupae injected with β−1,3-glucans

We employed a candidate gene approach to test if a β−1,3-glucan injection (above) elicits an immune response in pupae, with saline injected pupae as a control (*n* = 11, each for pupae frozen 0 hr and 48 hr after injection, for both saline and β-glucan treatments). Total RNA was extracted from pupae using the Maxwell RSC simply RNA tissue kit (Promega, Madison, Wisconsin) according to manufacturer's instructions, with a final elution volume of 60 µl. Reverse transcription was performed using the iScript cDNA synthesis kit (Bio-Rad, Hercules, California) as per the manufacturer's recommendations. Primer sequences were taken from (*Konrad et al., 2012*) or developed from cDNA sequence information of *L. neglectus* (*Table 4*). Gene expression analyses of 28S Ribosomal Protein S18a (used as housekeeping gene, which we had previously found to be stably expressed in pupae), Prophenoloxidase (*proPO*), Peptidoglycan Recognition Protein SC2 (*PGRP-SC2*) and β−1,3-glucan binding protein (*β−1,3-GBP*) were performed in 20 µl reaction volumes using KAPA SYBR Fast qPCR master mix (Kapa Biosystems, Wilmington, Massachusetts) and 0.2 µM each of specific primers (Sigma-Aldrich) on a Bio-rad CFX96 real-time PCR detection system. Two microliters of the cDNA sample were added per reaction and each sample was analysed in duplicate or triplicate wells. Each run contained an absolute negative as well as a no reverse transcription control. Primer efficiency was >95% for all primer sets using standard curves of 10-fold dilutions, and primer specificity was monitored based on a melting curve analysis following each run. We used the following program for amplification: 95°C for 5 min, followed by 40 cycles of 10 s of 95°C denaturation and 30 s of 60°C

**Table 4.** Primer information.

Primer sequences, annealing temperatures and amplicon sizes for the immune genes *proPO*, *PGRP-SC2* and *β−1,3-GBP* and the reference house keeping gene *28S RP S18a* of the invasive garden ant, *Lasius neglectus*, as obtained from (**Konrad et al., 2012**) and cDNA sequence information (Meghan L. Vyleta, AVG, SC unpublished data).

| Primer | Sequence | Amplicon length | Annealing temperature |
|---|---|---|---|
| *proPO* | F: 5'-TCTTTCTCGCGGTCTTGACT<br>R: 5'-TTGTTGGCGACGATTCTGTA | 99 bp | 60°C |
| *PGRP-SC2* | F: 5'-GTGGAGTGGATAACGGCGAA<br>R: 5'-CTATCTCCGGGACAGACGGT | 85 bp | 55°C |
| *β−1,3-GBP* | F: 5'-CTGCGCATATCAATTCCCGAC<br>R: 5'-TTCGCTATCTGTCCCGCTTC | 101 bp | 55°C |
| *28S RP S18a* | F: 5'-CGGCTGTATGCTACCACGTA<br>R: 5'-AAGCCTGCTTTCTGAGCCAT | 93 bp | 55°C |

DOI: https://doi.org/10.7554/eLife.32073.023

(55°C) annealing/extension. Normalised gene expression values (the average of technical replicates standardised to the housekeeping gene) were analysed using Mann-Whitney *U* tests and the resulting p values were corrected for multiple comparisons.

## Chemical analysis of β-glucan injected pupae

The chemical profiles of β−1,3-glucan and saline-injected pupae were analysed using GC–MS, following the above protocols. We tested whether the four CHCs that were increased specifically on unpacked pupae were also increased on pupae 48 hr after injection with either β-glucans or saline. We used LMERs to compare the CHC abundances within treatments, on pupae immediately after injection (saline, *n* = 27; β-glucan, *n* = 26) and 48 hr later (saline, *n* = 22; β-glucan, *n* = 22), correcting the p values for multiple comparisons across the CHCs. CHC abundances were square root transformed to a normal distribution. Since this experiment was carried out on 2 separate days, run was included as a random intercept effect to account for any potential, uncontrollable differences. The assumption of homogeneity of variances was violated for the LMER analysing Tritriacontadiene, though visual inspection of the model residuals found this violation to be relatively minor. Still, to test for the robustness of the LMER result, we also analysed Tritriacontadiene using a non-parametric test that does not make any assumptions about data distribution, but is unable to account for the random effect. This test also found a strong, significant difference between the two time points (Mann-Whitney *U* test, *U* = 157, p=0.007); hence we report the result of the LMER.

## Effect of destructive disinfection on pathogen replication

To test if destructive disinfection prevents *Metarhizium* from successfully replicating, we kept single pathogen-exposed pupae in petri dishes containing groups of 3 or 8 ants. This allowed us to assess how group size affects the likelihood of fungal inhibition. For the following 10 days, we observed the pupae for unpacking. When a pupa was unpacked, we left it with the ants for a further 1 or 5 days so that they could perform destructive disinfection. This allowed us to assess how the duration of destructive disinfection affects the likelihood of fungal inhibition. The destructively disinfected pupae were then removed and placed into petri dishes on damp filter paper at 23°C (8 ants 1 day and 5 days, *n* = 22 pupae each; 3 ants 1 and 5 days, *n* = 18 pupae each). We did not surface sterilise the pupae as this might have interfered with the destructive disinfection the ants had performed. Removed pupae were observed daily for *Metarhizium* sporulation for 30 days. To determine how many pupae sporulate in the absence of destructive disinfection, we kept pathogen-exposed pupae without ants as a control and recorded the number that sporulated for 30 d (*n* = 25). We compared the number of pupae that sporulated after 1 and 5 days and in the absence of ants using logistic regressions and Tukey post hoc comparisons, separately for the two ant group sizes (*Figure 3A*, *Figure 3—figure supplement 1*).

## In vitro investigation of destructive disinfection

We examined the individual effects of unpacking, biting and poison application on destructive disinfection by performing these behaviours in vitro. Pathogen-exposed pupae were initially kept with ants so that they could perform sanitary care. After 3 days, we removed the pupae and split them up into three groups: (i) pupae that we left cocooned, (ii) experimentally unpacked and (iii) experimentally unpacked and bitten. We simulated the damage the ants achieve through biting by damaging the pupal cuticle and removing their limbs with micro scissors. The pupae were then treated with either synthetic ant poison (60% formic acid and 2% acetic acid, in water; applied at a dose equivalent to what ants apply during destructive disinfection; *Figure 3—figure supplement 2*) or autoclaved distilled water as a control, using pressurised spray bottles (Lacor, Spain) to evenly coat the pupae in liquid. Spraying was carried out at a distance of 36 cm from the pupae and lasted for 1 s. The pupae were allowed to air dry for 5 min before being rolled over and sprayed again and allowed to dry a further 5 min. All pupae were then placed into separate petri dishes and monitored daily for *Metarhizium* sporulation (cocooned + poison, $n = 24$; unpacked + poison + biting, $n = 24$; all other treatments, $n = 25$). The number of pupae sporulating was analysed using a logistic regression with Firth's penalised likelihood, which offers a solution to the monotone likelihood caused by the complete absence of sporulation in one of the groups (R package 'brglm' [*Kosmidis, 2013*]). Pupal manipulation (cocooned/unpacked only/unpacked and bitten), chemical treatment (water or poison) and their interaction were included as main effects (*Figure 3B*, *Figure 3—figure supplement 3*). We followed up this analysis with Tukey post-hoc comparisons (*Table 3*).

## Comparing synthetic and ant poison spraying

We confirmed that synthetic poison spraying resulted in pupae receiving an amount of poison within the natural range that is applied by ants during destructive disinfection. Pupae taken from a stock colony were experimentally unpacked and sprayed with synthetic poison. We then measured their pH (all as above; $n = 21$). To test if synthetic poison spraying was similar to natural ant spraying, we compared their pH to pupae destructively disinfected by ants (data from *Figure 1—figure supplement 3*) using a Mann-Whitney *U* test (*Figure 3—figure supplement 2*). We adjusted the p value to correct for using this dataset twice (here and in *Figure 1—figure supplement 3*).

## The effect of the pupal cocoon on ant poison application

To test if the pupal cocoon limits the amount of the ants' poison that reaches the pupae inside, we took pupae from a stock colony and sprayed half with synthetic ant poison (as above; $n = 10$) and left the other half untreated ($n = 10$). We then unpacked these pupae and measured their pH (as above). As an additional control, we first experimentally unpacked pupae before spraying them with synthetic poison ($n = 10$). We analysed pH pupae using a KW test with post hoc comparisons (*Figure 3—figure supplement 4*).

## Disease transmission from infectious and destructively disinfected pupae

We tested the impact of destructive disinfection on disease transmission within groups of ants by keeping them with sporulating pupae or pupae that had been destructively disinfected. Infections were established in pupae (as above) and half were allowed to sporulate ($n = 11$), whilst the other half were experimentally destructively disinfected (as above; $n = 11$). Pupae were then kept individually with groups of five ants in mini-nests (cylindrical containers [Ø=90 mm] with a second, smaller chamber covered in red foil [Ø=33 mm]). Ant mortality was monitored daily for 30 days. Dead ants were removed, surface sterilised (as above) and observed for *Metarhizium* sporulation. The number of ants dying from *Metarhizium* infections in each treatment was compared using a logistic regression (*Figure 4A*). Mini-nest identity was included as a random intercept effect as ants from the same group are non-independent.

## Observations of destructive disinfection in another supercolonial population and related species

To confirm that our findings are not an idiosyncrasy of our specific study population, we tested whether the destructive disinfection observed in the Seva *L. neglectus* population is also found in

another, genetically distinct supercolonial population (*Ugelvig et al., 2008*) and a related (congeneric), non-supercolonial/non-invasive species, *Lasius niger*. We sampled hundreds of queens and many thousands of workers from a 300 m$^2$ area of *L. neglectus* in the botanical gardens in Jena, Germany (50°55'54.6"N 11°35'08.4"E). The studied *L. niger* colony was raised from a single founding queen collected after a natural mating flight in Harpenden, UK (51°48'48.9"N 0°22'51.5"W) and reared in the laboratory for 3 years, by which point it contained several hundred workers. To test if these ants also perform destructive disinfection, we kept two workers with single, pathogen-exposed or control-treated pupae (following the same protocols as above; Jena supercolony, n = 23 replicates per treatment; *L. niger*, n = 20 per treatment). We observed the ants on a daily basis to record the occurrence of destructive disinfection for 10 d. In both the Jena population and *L. niger*, no control-treated pupae were destructively disinfected (proportion ± 95% CIs: Jena = 0 ± 0–0.14; *L. niger* = 0 ± 0–0.16), whilst >60% of the pathogen-exposed pupae were destructively disinfected (proportion ± 95% CIs: Jena = 0.61 ± 0.41–0.78; *L. niger* = 0.95 ± 0.76–0.99).

## Acknowledgements

We thank L Lovicar for producing SEMs, B Leyrer and E Flechl for help with the chemical analysis and RNA extraction, respectively, B Milutinović and M Bračić for assistance with the chemical bioassay, and the *Social Immunity* group at IST Austria for ant collection and comments on earlier drafts of the manuscript. We are grateful to M Sixt, D Siekhaus and J J Boomsma for discussion of the project throughout. Finally, we thank R Rosengaus and an anonymous referee for their very helpful comments on an earlier version on the manuscript.

## Additional information

### Funding

| Funder | Grant reference number | Author |
| --- | --- | --- |
| European Research Council Seventh Framework Programme | ERC Starting Grant 243071 | Sylvia Cremer |
| European Research Council Seventh Framework Programme | MC-IEF 302004 | Line V Ugelvig |

The funders had no role in study design, data collection and interpretation, or the decision to submit the work for publication.

### Author contributions

Christopher D Pull, Conceptualization, Data curation, Investigation, Visualization, Methodology, Writing—original draft, Statistical analyses; Line V Ugelvig, Funding acquisition, Methodology, Writing—review and editing, Chemical analyses; Florian Wiesenhofer, Methodology, Writing—review and editing, Chemical analyses; Anna V Grasse, Methodology, Writing—review and editing, Gene expression analysis; Simon Tragust, Mark JF Brown, Conceptualization, Writing—review and editing; Thomas Schmitt, Writing—review and editing, Chemical analyses; Sylvia Cremer, Conceptualization, Supervision, Funding acquisition, Visualization, Writing—original draft, Project administration

### Author ORCIDs

Christopher D Pull http://orcid.org/0000-0003-1122-3982
Line V Ugelvig https://orcid.org/0000-0003-1832-8883
Sylvia Cremer https://orcid.org/0000-0002-2193-3868

### Ethics

Animal experimentation: The animal use protocol was performed in accordance with the IST Austria Ethics Committee. At present, the committee does not provide a specific approval numbers for

invertebrate animal research. Animals used in this study, Lasius neglectus, do not belong to regulated or protected species.

## Decision letter and Author response
Decision letter https://doi.org/10.7554/eLife.32073.028
Author response https://doi.org/10.7554/eLife.32073.029

## Additional files

### Supplementary files
• Transparent reporting form
DOI: https://doi.org/10.7554/eLife.32073.024

### Major datasets
The following dataset was generated:

| Author(s) | Year | Dataset title | Dataset URL | Database, license, and accessibility information |
|---|---|---|---|---|
| Pull C, Ugelvig L, Wiesenhofer F, Tragust S, Schmitt T, Brown M, Cremer S | 2017 | Destructive disinfection of infected brood prevents systemic disease spread in ant colonies | http://dx.doi.org/10.5061/dryad.73f45 | Available at Dryad Digital Repository under a CC0 Public Domain Dedication |

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
