## [Decision Letter]

[Editors’ note: a previous version of this study was rejected after peer review, but the authors submitted for reconsideration. The first decision letter after peer review is shown below.]

Thank you for submitting your work entitled "Destructive disinfection of infected brood prevents systemic disease spread in ant colonies" for consideration by *eLife*. Your article has been reviewed by two peer reviewers, and the evaluation has been overseen by a Reviewing Editor and a Senior Editor. The following individuals involved in review of your submission have agreed to reveal their identity: Rebeca Rosengaus (Reviewer #1).

Our decision has been reached after consultation between the reviewers. Based on these discussions and the individual reviews below, we regret to inform you that your work will not be considered further for publication in *eLife*.

The main point that led to the reject decision is not so evident from the text of the reviews, but became clear in the discussion session. Although the paper presents interesting and very careful work, it does not provide a big step forward in the field, in particular not in light of your previous publications on this topic. Although the destructive disinfection was considered to be novel, its exact mechanism is not very clear. As pointed out in review #1, the CHC profile would need to be further studied to ensure where it comes from. In fact, since this is the crucial signal, it would require deeper analysis to understand how it comes about. If it emanates indeed from the pupae, it would constitute an early phase immune response that would need to be clarified in more depth. In case you can substantially improve the paper on this aspect, you are welcome to submit a new version.

Reviewer #1:

This is an interesting manuscript in which the authors studied a previously unreported behaviour of "destructive disinfection" of ant pupae by Lasius workers. The authors report on serially linked behavioural acts that involve the removal of fungally infected pupae from its cocoon, followed by the perforation of the pupa's cuticle and finally the administration of poison secretions with antimicrobial properties. These coupled tasks ultimately reduce the risk of disease transmission/infection within a colony. The authors did a nice job in systematically addressing and tackling many of the aspects involved in this novel behaviour. Their methods, sample sizes and data analyses provide convincing evidence that this behaviour is inducible and adaptive. My major concern lays on the experimental design of the CHC. It appears the authors did not run chemical signature analyses on the triton+conidia suspension only? If this is the case, then they cannot ascertain that the changes in CHC profile of infected pupae represents "sickness cues" given off by the pupae. The possibility exists that those changes are of fungal (and not ant) origin. Given this and several other minor concerns, I cannot recommend this work for publication in its present form. Below some suggestions and hopefully constructive criticisms that I think may improve the clarity of their work. A pleasure reading such thorough research.

Reviewer #2:

Overview

Social insects have served for the past two decades as models to understand the evolution of immunocompetence (in the broad sense) in animal societies. Many social insect species face significant immune challenges due to their densely populated and large colonies in which disease transmission can be facilitated. Hymenopteran species may be genetically compromised due to haplodiploidy, and polyandrous mating may enhance worker disease resistance. Social immunity involves colony-level prophylactic response (sanitation) and/or the upregulation of immune factors related to vaccination-like effects resulting from social interactions, collectively reducing infection risk.

Current submission

In the present study, Lasius neglectus ants are described as using "destructive disinfection" to limit disease transmission of a fungal pathogen by removing pupae within which the pathogen is undergoing non-contagious incubation. Workers appear to be able to detect chemical cues emitted by pupae, recognizing disease risk and treating such pupae with antimicrobial secretions from the poison gland.

Evaluation

Colonies of social insects are known to identify, and sanitize, by allogrooming and/or antimicrobial secretion distribution, and/or remove infected larvae or workers, elevating colony temperature and generating "fever" and/or signalling the presence of pathogens following their detection. The authors' work extends our understanding of how individuals may serve immune-cell like functions to achieve colony-level resistance to disease. The finding that "the same principles of disease defence apply at different levels of biological organization" does not in and of itself appear to be unusual and/or highly novel and unexpected; it is a well-used superorganism analogy. As the authors note, previous studies have shown that workers (of several social insect species) remove infected brood. I do not see the response now reported to be substantially different in terms of how it advances our understanding of infection control in animal societies, although the authors coin a new term for their phenomenon. Overall, the "multicomponent behaviour' appears to be to remove infected individuals and sanitize them, killing them in the process (intentionally or unintentionally). This category of response to infection has been previously been shown in other social insects, for example, in the general context of undertaking behaviour. A more novel and significant result would be if workers did not destructively disinfect, but rather identified pre-contagious pupae and "medically" treated them with poison gland secretion, improving their survival.

[Editors’ note: what now follows is the decision letter after the authors submitted for further consideration.]

Thank you for resubmitting your work entitled "Destructive disinfection of infected brood prevents systemic disease spread in ant colonies" for further consideration at *eLife*. Your revised article has been favourably evaluated by Diethard Tautz (Reviewing editor) and two reviewers.

The reviewers have no further comments on the experimental side, but reviewer #2 (and after the discussion of these comments reviewer #1 agreed to them as well) suggests putting the results into a broader context. Please look through your text (i.e. Introduction/Discussion) and include these perspectives where it seems appropriate.

Reviewer #1:

The authors have addressed all my concerns and I believe that adding those extra experiments on the CHC has made this a robust and complete contribution to the field of social insect eco-immunology. I recommend this work for publication in its current form.

Reviewer #2:

In their revised manuscript, the authors have effectively responded to reviewer comments by adding new data and refining the narrative. The new studies add details absent in the initial submission that demonstrate chemicals used to disinfect brood are not of fungal origin, and workers thus detect cues associated with disease symptoms. Independent of the potential impact of the study in the broader analysis of social immunocompetence, the work is rigorous, advances our understanding of mechanisms underpinning social responses to infection control, and is to be applauded. However, the question remains as to whether adding details of the mechanisms regulating the disinfection process in ants is broadly significant or serves a more specialized audience. Here it can also be noted that the first sentence of the revised Abstract focuses the study in social insect biology, rather than the broader ecological immunology of animal social systems, and "completing the (superorganism) analogy" similarly accentuates insect sociobiology. My sense is that most practitioners in the field would agree that "the same principles of disease defence apply at different levels of biological organization" is already understood, at the level of analogy. The authors draw parallels and note variation among honey bee and ant systems of disease control. The more general message for the analysis of social and ecological immunology should be clarified. For example, aside from the mechanistic details, do the present results offer deeper insights into the concept of herd immunity, applied across diverse taxa? If so, the authors should state so clearly, as it this would be more informative than "completing the superorganism analogy."

---

## [Author Response]

[Editors’ note: the author responses to the first round of peer review follow.]

The main point that led to the reject decision is not so evident from the text of the reviews, but became clear in the discussion session. Although the paper presents interesting and very careful work, it does not provide a big step forward in the field, in particular not in light of your previous publications on this topic. Although the destructive disinfection was considered to be novel, its exact mechanism is not very clear. As pointed out in review #1, the CHC profile would need to be further studied to ensure where it comes from. In fact, since this is the crucial signal, it would require deeper analysis to understand how it comes about. If it emanates indeed from the pupae, it would constitute an early phase immune response that would need to be clarified in more depth. In case you can substantially improve the paper on this aspect, you are welcome to submit a new version.

We have performed additional experiments and chemical analyses that reveal the chemical changes observed in infected, unpacked pupae are not of fungal origin and can be similarly elicited by an experimental immune challenge with fungal cell wall material, which induces both an immune reaction and a change in the cuticular hydrocarbon profile. We have addressed this point using separate approaches that have produced complimentary results.

Firstly, we used GC–MS to analyse the chemical profiles of all of the separate life stages of the fungus: the conidiospores, blastospores and mycelia. By comparing the chemical profiles of the fungus to those of the pupae, we find that there are no peaks of fungal origin in the pupae profiles. Hence, this analysis provides strong evidence that the changes in the CHC profile of infected, unpacked pupae are indeed of pupal origin.

Secondly, we injected pupae with a fungal immune elicitor, β-1,3-glucans, which act as a major pathogen-associated molecular pattern (PAMP) for the invertebrate immune system. β-1,3-glucans are highly conserved components of fungal cell walls, including *Metarhizium*, known to trigger the major antimicrobial immune defences of invertebrates. Hence, they can be used to elicit an immune reaction in the absence of an actual pathogen. Using qPCR, we found that β-glucan injected pupae had increased expression of immune genes; namely, an IMD pathway regulator gene and a β-glucan binding protein gene that specifically recognises fungal cell wall components. Injection of fungal PAMPs also altered the chemical profile of the pupae, resulting in an increase in two out of the four CHCs that were also in higher abundance on *Metarhizium*-infected, unpacked pupae. Interestingly, these are also the same type of CHCs that are increased on sick honeybees, again suggesting that they may act as conserved “sickness cues” within the social Hymenoptera.

We have added the fungal chemical analysis to the manuscript:

1) Description of the fungal life stages in the Introduction: “Penetration is followed by a short, non-infectious incubation period of a few days, during which single cell fungal blastospores produce toxins that suppress the host’s immune system and eventually cause host death. The fungus then enters a second, non-infectious mycelial stage that grows saprophytically throughout the corpse. Once all available nutrients have been consumed by the mycelia, the fungus grows out of the corpse and releases millions of new infectious conidiospores in a process called sporulation, about 1-3 days after host death [Hughes, Eilenberg and Boomsma, 2002]”.

2) The methodology for the cultivation of the different fungal life stages: “In addition to fungal conidiospores, we also cultured blastospores and mycelia to obtain the chemical profiles (see below) of all stages of the fungi’s lifecycle [Deacon, 2006]. […] We incubated the flask (5 days, 180 rpm, 27°C) and vacuum-filtered (40µm mesh; Millipore Steriflip) the resulting fungal mass to remove the liquid broth. We then washed the mycelial mass three times in autoclaved distilled water”.

3) The chemical analysis and comparison to pupae: “Comparison of fungal and pupal chemical profiles. One millilitre aliquots of conidiospore (10^9^/ml in 0.05% TX) and blastospore (4x10^6^/ml in PBS) suspensions and approx. 500 mg of mycelia (in 500 µl of autoclaved distilled water) were washed three times by briefly vortexing and centrifuging the samples (5 min, 5500 g), discarding the supernatants, and replacing with 1 ml of autoclaved distilled water for the first two washes, and 500 µl for the last wash. […] This analysis revealed that none of the fungal compound RIs were overlapping with the pupal compounds, hence confirming that the identified pupal substances are not of fungal origin”.

4) And added in new results: “There were no novel compounds present on unpacked or cocooned pathogen-exposed pupae that were not also present on control pupae (Table 1; Figure 2—figure supplement 2), suggesting that these differences were not caused by odours emitted directly by the fungus, but were of pupal origin. By analysing the chemical profiles of each of the pathogen’s separate developmental stages (infectious condiospores, post-infection blastospores, and saprophytic mycelium) and performing a direct comparison of the fungal compounds to the pupal chemical profiles, we confirmed that there were no fungus-derived peaks in the pupal profiles (see Materials and methods for more information)”.

We have added the experimental injection of the fungal cell wall stimulus and its effect on the pupal chemical profile and immune gene expression:

1) The results of this experiment: “To investigate the possibility that CHC changes on unpacked pupae are the result of an immune response developed by the host, we injected pupae with β-1,3-glucans – polysaccharides that are an integral component of fungal cell walls, including Metarhizium. […]These data reveal that some of the changes in pupal chemical profile can be directly linked to a host reaction to an immune elicitor, similar to findings in honeybees [Richard, Holt and Grozinger, 2012; Richard, Aubert and Grozinger, 2008], mice [Arakawa, Cruz and Deak, 2011] and humans [Shirasu and Touhara, 2011]”.

2) The methodology for this experiment: “Immune stimulation of pupae using β-glucans. We injected β-1,3-glucans to test whether the changes in the chemical profile of infected pupae may be caused by an immune stimulation [Vilcinskas and Wedde, 1997; Unestam and Söderhäll, 1977; Gunnarsson, 1988]. […] This test also found a strong, significant difference between the two time points (Mann-Whitney U test, U = 157, p = 0.007); hence we report the result of the LMER”.

3) Two new supporting figures and legends: “Figure 2—figure supplement 3 and Figure 2—figure supplement 4.

Reviewer #1:This is an interesting manuscript in which the authors studied a previously unreported behaviour of "destructive disinfection" of ant pupae by Lasius workers. The authors report on serially linked behavioural acts that involve the removal of fungally infected pupae from its cocoon, followed by the perforation of the pupa's cuticle and finally the administration of poison secretions with antimicrobial properties. These coupled tasks ultimately reduce the risk of disease transmission/infection within a colony. The authors did a nice job in systematically addressing and tackling many of the aspects involved in this novel behaviour. Their methods, sample sizes and data analyses provide convincing evidence that this behaviour is inducible and adaptive. My major concern lays on the experimental design of the CHC. It appears the authors did not run chemical signature analyses on the triton+conidia suspension only? If this is the case, then they cannot ascertain that the changes in CHC profile of infected pupae represents "sickness cues" given off by the pupae. The possibility exists that those changes are of fungal (and not ant) origin. Given this and several other minor concerns, I cannot recommend this work for publication in its present form. Below some suggestions and hopefully constructive criticisms that I think may improve the clarity of their work. A pleasure reading such thorough research.

As detailed above, we can now show that the chemical changes are not of fungal origin, having analysed the chemical profiles of all the fungi’s developmental stages and comparing them to those of infected, unpacked pupae. Instead, we can show that these changes are ant-derived and occur in response to an immune stimulation caused by fungal material inside their body.

Reviewer #2:OverviewSocial insects have served for the past two decades as models to understand the evolution of immunocompetence (in the broad sense) in animal societies. Many social insect species face significant immune challenges due to their densely populated and large colonies in which disease transmission can be facilitated. Hymenopteran species may be genetically compromised due to haplodiploidy, and polyandrous mating may enhance worker disease resistance. Social immunity involves colony-level prophylactic response (sanitation) and/or the upregulation of immune factors related to vaccination-like effects resulting from social interactions, collectively reducing infection risk.Current submissionIn the present study, Lasius neglectus ants are described as using "destructive disinfection" to limit disease transmission of a fungal pathogen by removing pupae within which the pathogen is undergoing non-contagious incubation. Workers appear to be able to detect chemical cues emitted by pupae, recognizing disease risk and treating such pupae with antimicrobial secretions from the poison gland.EvaluationColonies of social insects are known to identify, and sanitize, by allogrooming and/or antimicrobial secretion distribution, and/or remove infected larvae or workers, elevating colony temperature and generating "fever" and/or signalling the presence of pathogens following their detection. The authors' work extends our understanding of how individuals may serve immune-cell like functions to achieve colony-level resistance to disease. The finding that "the same principles of disease defence apply at different levels of biological organization" does not in and of itself appear to be unusual and/or highly novel and unexpected; it is a well-used superorganism analogy. As the authors note, previous studies have shown that workers (of several social insect species) remove infected brood. I do not see the response now reported to be substantially different in terms of how it advances our understanding of infection control in animal societies, although the authors coin a new term for their phenomenon. Overall, the "multicomponent behavior' appears to be to remove infected individuals and sanitize them, killing them in the process (intentionally or unintentionally). This category of response to infection has been previously been shown in other social insects, for example, in the general context of undertaking behavior. A more novel and significant result would be if workers did not destructively disinfect, but rather identified pre-contagious pupae and "medically" treated them with poison gland secretion, improving their survival.

We agree that the concept of the supercolony analogy of social immunity has been established in a previous review (Cremer and Sixt, 2009). Yet we feel that the current work provides an experimental proof of concept that substantially extends the previously reported removal of infected brood (“hygienic behaviour”), as it includes a novel destruction phase, which is required to kill the pathogen. Hence, only this last step completes the analogy and also reveals very similar mechanisms to the find me/eat me signal emitted by infected cells, and the cell membrane piercing by perforins from the immune cells, followed by the injection of cytotoxins. As discussed, conventional hygienic behaviour, i.e. only the removal of infected brood in the absence of destructive disinfection, seems sufficient in honeybees, as they drop the infected brood to the ground, where they will not be re-encountered and hence the risk of re-infection is minimal. Ants, on the contrary, will re-encounter the dead they remove in the ‘two-dimensional’ territory surrounding their nest, hence potentially applying a stronger selection pressure for them to evolve effective mechanisms to prevent pathogen replication and re-infection of the colony. However, existing studies (Ugelvig et al. 2010, Tragust et al. 2013) failed to observe ant handling of the infected brood after its removal. In both honeybees and ants, these behaviours occur before host death, and are performed in addition to regular undertaking behaviours, such as removing and sometimes burrowing the dead (though, to our knowledge, disinfection has never been reported as a component of undertaking behaviour so far; see Sun and Zhou, 2013 for a recent review). As detailed in the third paragraph of the Introduction, we previously described that ants use their poison as a sanitary treatment, which acts as a first line of defence to prevent contaminated pupae from contracting infections (Tragust et al. 2013). Here, we describe a second, distinct line of defence, which requires the use of poison in a more complex, combination of behaviours, when this first line of defence fails. We therefore feel that our work is a significant advancement for the field, given that few animals are known to actively prevent successful infections from developing and transmitting to new hosts; generally infected individuals are simply avoided.

[Editors' note: the author responses to the re-review follow.]

The reviewers have no further comments on the experimental side, but reviewer #2 (and after the discussion of these comments reviewer #1 agreed to them as well) suggests putting the results into a broader context. Please look through your text (i.e. Introduction/Discussion) and include these perspectives where it seems appropriate.

We agree that the previous Introduction and Discussion of our work was very social insect focused. We have now added text to the Introduction and Discussion to make it broader and to place it into the established conceptual framework of ecological immunology. We have also amended our discussion to make the relations between our findings and those in other social systems more explicit.

Reviewer #1:The authors have addressed all my concerns and I believe that adding those extra experiments on the CHC has made this a robust and complete contribution to the field of social insect eco-immunology. I recommend this work for publication in its current form.

We thank the reviewer very much for their positive appraisal of our work and additional findings.

Reviewer #2:In their revised manuscript, the authors have effectively responded to reviewer comments by adding new data and refining the narrative. The new studies add details absent in the initial submission that demonstrate chemicals used to disinfect brood are not of fungal origin, and workers thus detect cues associated with disease symptoms. Independent of the potential impact of the study in the broader analysis of social immunocompetence, the work is rigorous, advances our understanding of mechanisms underpinning social responses to infection control, and is to be applauded. However, the question remains as to whether adding details of the mechanisms regulating the disinfection process in ants is broadly significant or serves a more specialized audience. Here it can also be noted that the first sentence of the revised Abstract focuses the study in social insect biology, rather than the broader ecological immunology of animal social systems, and "completing the (superorganism) analogy" similarly accentuates insect sociobiology. My sense is that most practitioners in the field would agree that "the same principles of disease defence apply at different levels of biological organization" is already understood, at the level of analogy. The authors draw parallels and note variation among honey bee and ant systems of disease control. The more general message for the analysis of social and ecological immunology should be clarified. For example, aside from the mechanistic details, do the present results offer deeper insights into the concept of herd immunity, applied across diverse taxa? If so, the authors should state so clearly, as it this would be more informative than "completing the superorganism analogy."

We thank the reviewer for directing us to the importance of the broader ecological immunology framework, and agree that the previous version was indeed too focused on social insects, likely hiding the relevance of our findings for the wider field as a whole. In the new Abstract, Introduction and Discussion, we compare and contrast our findings in ants to other group-living animals, but make explicit why some social insects are different from other social groups, given that they have made the transition to superorganismality. We consider this an important addition, as it highlights that (i) defences such as destructive disinfection are likely to only evolve in superorganisms, due to workers typically gaining all their fitness indirectly, and (ii) that this should promote the evolution of explicit sickness signalling by infected group members, as we find in our study. In this new version we hence discuss our work much more conceptually, which we hope proves interesting for both specialists and non-specialists alike.